# CONCISE REASONING IN THE LENS OF LAGRANGIAN OPTIMIZATION

## ABSTRACT

Concise reasoning in large language models seeks to generate only essential intermediate steps needed to arrive at a final answer, thereby alleviating issues of "overthinking". Most proposed approaches hinge on carefully hand-crafted heuristics, struggling to balance concision with performance, often failing to adapt across domains and model scales. In this work, we address these challenges by introducing a principled and pragmatic strategy, performance-aware length updating (PALU). As a principled algorithm, PALU formulates concise reasoning as a constrained optimization problem, minimizing response length subject to a performance constraint, and then applies *Lagrangian* optimization to convert it into a tractable unconstrained problem. As a pragmatic solution, PALU streamlines complicated update rules through three approximations: *(i)* estimating performance with off-policy rollouts, *(ii)* truncating the *Lagrange* multiplier to two extremes, and *(iii)* replacing gradient-based updates with quantile-driven length adjustments. PALU reduces output length by 64% while improving accuracy by 16% when applied to DEEPSEEK-DISTILL-QWEN-1.5B, averaged over six benchmarks, outperforming a range of alternative methods. Furthermore, PALU is demonstrated to adapt across both domain (logic, STEM and math) and model scale (1.5B, 7B, 14B) entrenching the algorithm as a practical and effective concise reasoning approach.

## 1 INTRODUCTION

Reasoning, requiring large language models (LLMs) to work through intermediate steps before producing a final answer, substantially improves performance on complex tasks such as mathematics (Jaech et al., 2024; Shao et al., 2024), programming (Lambert et al., 2024), and value alignment (Guo et al., 2025). Yet this benefit is often accompanied by overthinking: redundant self-reflection, backtracking, and validation (Chen et al., 2024; Zhang et al., 2024; Fatemi et al., 2025). These limitations inflate inference costs and hampers user experience, motivating the need for *concise reasoning*—the production of only the essential steps required to reach a correct answer.

Reinforcement learning (RL), with its proven success in incentivizing LLM reasoning ability (Guo et al., 2025; Jaech et al., 2024), emerges as a natural and mature avenue toward concise reasoning. Existing RL-based concise reasoning solutions typically either *(i)* employ carefully shaped reward functions to discourage overlong generations (Xiang et al., 2025; Yeo et al., 2025; Chen et al., 2025) or *(ii)* impose rigid length budgets that truncate overthinking trajectories (Hammoud et al., 2025; Hou et al., 2025) during the training. These heuristic attempts, albeit promising, implicitly set a target generation length for dataset queries globally or individually, and then penalize or discard the generations with length exceeding this pre-defined value. Consequently, they often demand extensive human effort to adapt across domains and model scales, and struggle to balance conciseness with performance because of the sole conciseness objective. This raises a research question:

*Can we achieve concise reasoning that (i) balances performance with conciseness, (ii) adapts across domains and model sizes without re-tuning, and (iii) avoids increases in training compute?*

In this work, we address this challenge by introducing performance-aware length update (PALU), an algorithmic strategy that adaptively updates the LLMs token generation budget to achieve a state of conciseness without sacrificing accuracy and to generalize across diverse domains and model scales.

As a principled strategy, PALU formulates concise reasoning as a constrained optimization problem: minimize rollout length while maintaining performance above a specified threshold. Because constrained problems are difficult to solve directly, PALU adopts a *Lagrangian* formulation that converts the constraint into an equivalent unconstrained objective. An associated *Lagrange* multiplier then dynamically balances concision and performance, yielding PALU's first key property: concise reasoning without hand-tuned length heuristics while maintaining performance.

As a pragmatic solution, PALU replaces expensive min–max gradient updates for the *Lagrangian* with three practical approximations.

(i) Off-policy performance check. Instead of collecting fresh rollouts to determine the *Lagrange* multiplier update direction, PALU reuses last-epoch rollouts to estimate performance. This avoids repeated model loading and new rollout computation, thereby preserving *Efficiency*.

(ii) Regime-based optimization scheme. Rather than tuning the *Lagrange* multiplier via brittle, slow ascent, PALU snaps the multiplier into two extremes implicitly. This simplification preserves the essential sign behavior of $\lambda$ and ensures conciseness without compromising performance, yielding *Balance*.

(iii) Quantile-driven budget update. Because gradients of the *Lagrangian* with respect to the length budget are non-differentiable, PALU uses a quantile-based surrogate: it estimates the marginal effect of reducing the budget by observing accuracy drops and sets the step size by a target quantile of these drops. Grounded in these derivative-inspired statistics, the update scales naturally across domains and model sizes without heuristic retuning, conferring *Adaptivity*,

PALU, when combined with GRPO (Shao et al., 2024), reduces generation length by 64% while improving accuracy by 16% on DEEPSEEK-R1-DISTILL-QWEN-1.5B, averaged across six benchmark tasks, outperforming alternative methods. Compared with methods that rely on heuristic length budgets or length-aware rewards, both of which require sensitive tuning across domains and model sizes, PALU achieves superior conciseness across multiple domains (logic, STEM, mathematics) and scales effectively from 1.5B to 14B parameters. By uniting conciseness with performance, and exhibiting strong adaptivity across domains and scales, PALU demonstrates the effectiveness of a principled yet pragmatic solution for concise reasoning.

## 2 PRELIMINARIES

Group Relative Policy Optimization (GRPO (Shao et al., 2024)) simplifies PPO (Schulman et al., 2017) for LLM finetuning by replacing the heavy value model with a per-prompt, group-relative normalization of the reward. Specifically, given a question–answer pair $(q, a)$ drawn from dataset $\mathcal{D}$, a group of $G$ rollouts (responses) $\{o_i\}_{i=1}^{G}$ is sampled, and their advantages are computed as:

$$\hat{A}_i(o_i, a) = \frac{r(o_i, a) - \text{mean}(\{r(o_i, a)\}_{i=1}^{G})}{\text{std}(\{r(o_i, a)\}_{i=1}^{G})}, \tag{1}$$

where the reward signal $r$ is provided by some rule-based reward functions. To stabilize training, GRPO adopts the clipped surrogate objective from PPO (Schulman et al., 2017):

$$\min\left\{ r_{i,t}(\boldsymbol{\theta})\hat{A}_i(o_i, a), \ \text{clip}\left(r_{i,t}(\boldsymbol{\theta}), 1 - \epsilon_{\text{low}}, 1 + \epsilon_{\text{high}}\right) \right\}, \tag{2}$$

where $r_{i,t}(\boldsymbol{\theta})$ is the per-token probability ratio between policy $\pi_{\boldsymbol{\theta}}$ and the behavior policy $\pi_{\boldsymbol{\theta}_{\text{old}}}$:

$$r_{i,t}(\boldsymbol{\theta}) = \frac{\pi_{\boldsymbol{\theta}}(o_{i,t} \mid q, o_{i,<t})}{\pi_{\boldsymbol{\theta}_{\text{old}}}(o_{i,t} \mid q, o_{i,<t})}. \tag{3}$$

This yields the GRPO objective (we eliminate the KL-divergence constraint (Yu et al., 2025)):

$$J_{\text{GRPO}}(\pi_{\boldsymbol{\theta}}) = \mathbb{E}_{(q,a)\sim\mathcal{D}, \{o_i\}_{i=1}^{G}\sim\pi_{\boldsymbol{\theta}_{\text{old}}}(\cdot|q,L)}$$
$$\left[ \frac{1}{G}\sum_{i=1}^{G} \frac{1}{|o_i|}\sum_{t=1}^{|o_i|} \min\left\{ r_{i,t}(\boldsymbol{\theta})\,\hat{A}_i(o_i, a), \ \text{clip}(r_{i,t}(\boldsymbol{\theta}), 1 - \epsilon_{\text{low}}, 1 + \epsilon_{\text{high}})\,\hat{A}_i(o_i, a) \right\} \right], \tag{4}$$

where $L$ denotes the length budget for generation, such that decoding proceeds token by token and is forcibly terminated once the number of generated tokens reaches $L$.

## 3 RELATED WORK

Concise reasoning in LLMs is an emerging research direction aimed at mitigating the overthinking phenomenon (Han et al., 2024; Ma et al., 2025). Existing solutions can be broadly categorized into three paradigms: *(i)* training-free methods, including guided prompting (Xu et al., 2025b), modular workflow pipelines (She et al., 2025), decoding manipulation (Muennighoff et al., 2025), and latent-space reasoning (Hao et al., 2024); *(ii)* SFT- and DPO-based methods, including reasoning path filtering (Munkhbat et al., 2025), reasoning with latent tokens (Su et al., 2025), and preference optimization (Team et al., 2025a); and *(iii)* RL-based methods, to which our approach belongs.

Table 1: An overview of RL-based concise reasoning methods.

| Modification | Penalty function | Representatives |
|---|---|---|
| Reward function | $r = r(o, a) - f(\text{len}(o))$ | Kimi 1.5 RL (Team et al., 2025a); Overlong punishment (Yu et al., 2025) |
| Reward function | $r = r(o, a) - f(\text{len}(o), \text{diff}(q))$ | L1 (Aggarwal & Welleck, 2025), |
| Reward function | $r = r(o, a) - f(\text{len}(o) - \text{target})$ | O1-pruner (Luo et al., 2025b); ShorterBetter (Yi et al., 2025) |
| Length budget | $L = f(\text{stage})$ | Thinkprune (Hou et al., 2025) |
| Length budget | $L = f(\text{diff}(q))$ | GFPO (Shrivastava et al., 2025) |

Reward-function-based approaches typically introduce length-aware penalties during RL training. Team et al. (2025a); Xiang et al. (2025); Arora & Zanette (2025); Yeo et al. (2025); Song & Zheng (2025) subtract a penalty term proportional to response length from reward signals. Others (Xiang et al., 2025; Shen et al., 2025; Li et al., 2025) refine this idea by incorporating both response length and question difficulty. A further refinement discounts the reward according to the deviation between the generated and the target length (Luo et al., 2025b; Yi et al., 2025; Team et al., 2025b). However, aggregating such heterogeneous reward components prior to normalization can distort the length penalty (Chen et al., 2025). Moreover, these methods face a fundamental limitation in adaptivity: their reward shapes require extensive trial-and-error tuning across data domains and model scales.

Length-budgeting methods, by contrast, regulate the rollout through setting hard length budgets. This approach would stop the decoding when the number of generated tokens reaches this value. One line of work (Hou et al., 2025; Hammoud et al., 2025) progressively reduces the global length budget, whereas another (Shrivastava et al., 2025) filters trajectories after generation, retaining only those shorter than a length threshold. A limitation of these approaches is that the budget is typically set heuristically, often neglecting the risk of performance degradation. Our method instead allocates the budget in a principled manner, explicitly balancing conciseness with performance. For a more comprehensive survey on concise reasoning methods, please refer to Zhu & Li (2025).

## 4 PROPOSED METHOD: PALU

### 4.1 FORMULATION AND INTUITION

Unlike heuristic approaches, we formulate concise reasoning into a constrained optimization problem. Let $L$ denote the per-question length budget, $r$ a (rule-based) reward evaluating the responses from a reasoning model $\pi_{\boldsymbol{\theta}}$, and $C \in [0, 1]$ a global performance threshold. The objective is to minimize $L$ while ensuring performance meets or exceeds $C$ for question-answer pairs $\{(q, a)\}$ drawn from dataset $\mathcal{D}$:

$$\min_{\boldsymbol{\theta}, L>0} L \quad \text{s.t.} \quad R(\boldsymbol{\theta}, L, q) \geq C, \tag{5}$$

with $R(\boldsymbol{\theta}, L, q)$ denoting the expected reward obtained by model $\pi_{\boldsymbol{\theta}}$, when generating a set of response $\boldsymbol{o}$ for query $q$ under a length budget $L$:

$$R(\boldsymbol{\theta}, L, q) = \mathbb{E}_{\boldsymbol{o} \sim \pi_{\boldsymbol{\theta}}(\cdot \mid q, L)} \big[ R(\boldsymbol{o}, a) \big]. \tag{6}$$

Directly solving Eq. (5) can be difficult. Fortunately, *Lagrangian* optimization enables a conversion of the original problem to the following min–max objective:

$$\min_{\boldsymbol{\theta}, L>0} \max_{\lambda \geq 0} \mathcal{L}(\boldsymbol{\theta}, L, \lambda) = L + \lambda \Big( C - R(\boldsymbol{\theta}, L, q) \Big), \tag{7}$$

where $\lambda$ is the dual variable penalizing constraint violation. Assuming differentiability, the solution of the original constrained optimization can be approximated by applying first-order stochastic updates with learning rates $\eta_\lambda$, $\eta_\theta$, and $\eta_L$ (for the dual variable, model parameters, and length budget, respectively), together with implicit projections onto $\lambda \geq 0$ and $L > 0$:

$$\lambda \leftarrow \lambda + \eta_\lambda \Big( C - R(\boldsymbol{\theta}, L, q) \Big), \tag{8}$$

$$\boldsymbol{\theta} \leftarrow \boldsymbol{\theta} + \eta_\theta \cdot \lambda \cdot \nabla_{\boldsymbol{\theta}} R(\boldsymbol{\theta}, L, q), \tag{9}$$

$$L \leftarrow L - \eta_L \Big( 1 - \lambda \cdot \nabla_L R(\boldsymbol{\theta}, L, q) \Big). \tag{10}$$

These updates admit a natural interpretation. When the performance constraint is satisfied, $\lambda$ remains small and the corresponding length budget $L$ is reduced. Empirically, longer responses tend to correlate with higher reward, so $\nabla_L R \geq 0$. Conversely, when performance falls below $C$, $\lambda$ increases, expanding $L$ and prioritizing updates to $\boldsymbol{\theta}$ to restore performance. Beside the explicit balance between performance and conciseness, the update rule for length budget $L$, Eq. (10), offers a principled way to achieve the concise reasoning, without heuristics on the target generation length.

## 4.2 PRACTICAL ALGORITHM

Guided by the min–max formulation and the first-order update rules, we introduce performance-aware length update (PALU), a pragmatic and principled algorithm for training concise reasoning models. PALU simplifies the complicated updates rules by three components: *(i)* an off-policy pass-rate estimate, *(ii)* a regime-based optimization scheme that toggles the optimization focus, and *(iii)* a quantile-based surrogate for the derivative term $\nabla_L R(\boldsymbol{\theta}, L, q)$.

**Off-policy performance estimation (Eq. (8))**  Updating the length budget $L$ and model parameters $\boldsymbol{\theta}$ requires estimating the performance $R$. Computing this quantity on-policy would demand repeatedly reloading the latest parameters, which is computationally costly. Instead, we approximate it with the previous round's evaluation:

$$R(\boldsymbol{\theta}, L, q) \approx R\big(\boldsymbol{\theta}_{\text{old}}, L_{\text{old}}, q\big) = \mathbb{E}_{o \sim \pi_{\boldsymbol{\theta}_{\text{old}}}(\cdot \mid q, L_{\text{old}})}\big[ r(o, a) \big]. \tag{11}$$

This off-policy reuse provides a conservative estimate of the true pass rate. While such approximations are often unstable in reinforcement learning with randomly initialized policies, LLM fine-tuning differs because performance typically improves monotonically thanks to pretraining. Thus, this conservative bias is acceptable, and even desirable, because it naturally underestimates model performance and emphasizes more on policy improvement (Eq. (9), *i.e.*, the case of large $\lambda$).

**Regime-based optimization (Eq. (8) and Eq. (10))**  In the *Lagrangian* view, $\lambda$ reweights the emphasis between conciseness and performance. When the performance constraint is satisfied ($C - R \leq 0$), residuals integrate to a small $\lambda$, so the update prioritizes reducing the length budget. In this case, $\big(1 - \lambda \cdot \nabla_L R(\boldsymbol{\theta}, L, q)\big) > 0$. Conversely, when the constraint is violated ($C - R > 0$), a sequence of positive residuals drives $\lambda$ upward, shifting the emphasis toward recovering performance by enlarging $L$ and updating the model $\pi_\theta$.

While this continuous adjustment is elegant in theory, it depends critically on carefully tuned learning rates and a long integration horizon. Both impractical for LLM post-training. PALU therefore discards the need for a continuously evolving $\lambda$ and instead approximates only its sign behavior with a two-regime controller:

$$\text{Optimization regime} = \begin{cases} L \leftarrow L - \alpha_\tau^q & \text{if } R(\boldsymbol{\theta}, L, q) \geq C, \\ L \leftarrow L_{\max} & \text{otherwise} \end{cases}, \tag{12}$$

where $\alpha_\tau^q > 0$ is a new term we will explain later. This simplification turns the *Lagrange* multiplier into an implicit "bang–bang" controller with two regimes: one regime pushes toward conciseness, the other safeguards performance by resetting to the maximum $L_{\max}$ when the constraint is violated.

---

**Algorithm 1 Performance-Aware Length Update (PALU) with GRPO**

---

**Input:** initial model $\pi_{\boldsymbol{\theta}}$, dataset $\mathcal{D}$, bound $L_{\max}$, performance threshold $C$

1: **for** epoch in range($N$) **do**
2:     **for** each mini-batch $\mathcal{D}_b \subset \mathcal{D}$ **do**
3:         **if** first epoch **then**
4:             Initialize the length budget for all questions: $L = L_{\max}$
5:         **else**
6:             Reuse the last round pass rate, *e.g.,* Eq. (11)
7:             Update $L$ for each $q \in \mathcal{D}_b$ using rule Eq. (12)
8:         **end if**
9:         Collect responses $\boldsymbol{o}$ with parameter $\boldsymbol{\theta}$ and per sample budget $L$
10:        Update $\boldsymbol{\theta}$ with GRPO as per Eq. (4)
11:     **end for**
12: **end for**
13: **Output:** concise reasoning model $\pi_{\boldsymbol{\theta}}$

---

**Quantile-driven budget update (Eq. (10))** To set the per-question reduction step $\alpha_\tau^{(q)}$ used by the regime controller (Eq. (12)), we use Eq. (10) as an interpretive guide. The term $\nabla_L R(\boldsymbol{\theta}, L, q)$ captures the sensitivity of performance to the length budget. Because $R$ is a non-differentiable, rule-based reward, we approximate this sensitivity via the difference between two nearby operating points in the distribution of correct response lengths. Let

$$Q_\tau^{(q)} := \text{Quantile}_\tau \left( \{\text{len}(o_i)\}_{i=1}^G \mid o \sim \pi_{\boldsymbol{\theta}_{\text{old}}}(\cdot | q, L_{\text{old}}); \, r(o, a) = 1 \right) \qquad (13)$$

and define the quantile gap

$$\alpha_\tau^{(q)} := Q_{1.0}^{(q)} - Q_{1.0-\tau}^{(q)}. \qquad (14)$$

If $L$ is near $Q_{1.0}^{(q)}$, typical when the performance threshold $C$ is high, reducing $L$ by $\alpha_\tau^q$ lowers the success rate by approximately $\tau$. Hence,

$$\nabla_L R(\boldsymbol{\theta}, L, q) \approx \frac{R(\boldsymbol{\theta}, L, q) - R(\boldsymbol{\theta}, L - \alpha_\tau^{(q)}, q)}{\alpha_\tau^{(q)}} \;=\; \frac{\tau}{\alpha_\tau^{(q)}}, \qquad (15)$$

Substituting into Eq. (10) yields the budget update:

$$L = L - \eta_L \cdot \Delta L, \quad \Delta L = \left(1 - \lambda \cdot \nabla_L R(\boldsymbol{\theta}, L, q)\right) \approx \left(1 - \lambda \cdot \frac{\tau}{\alpha_\tau^{(q)}}\right) \propto \alpha_\tau^{(q)}. \qquad (16)$$

Accordingly, our regime update uses $L \leftarrow L - \alpha_\tau^{(q)}$ when $R(\boldsymbol{\theta}, L, q) \geq C$, with $\alpha_\tau^{(q)}$ as the gap between the longest correct response and its $(1-\tau)$-quantile length, capturing how dispersed correct responses are. In simple terms, when correct responses cluster tightly in length (small $\alpha_\tau$), updates proceed cautiously; when they exhibit a wider tail, the adjustment is correspondingly more aggressive. The resulting update embodies a direct, data-driven proxy for inverse sensitivity (the derivative term in Eq. (10)), capturing the essence of Lagrangian optimization within a pragmatic rule.

**Summary** PALU circumvents the instability and cost of the full *Lagrangian* multiplier method while retaining its principled grounding, by combining off-policy performance check, the regime-based controller, and the quantile-driven update step. This design offers three key advantages:

*(i) Efficiency*, no additional computations are required to estimate the performance,

*(ii) Balance*, the two-regime controller reconciles conciseness and performance,

*(iii) Adaptivity*, the quantile-based step scales naturally across domains and model scales.

Algorithm 1 presents the pseudocode of PALU, instantiated with the GRPO performance objective (Shao et al., 2024), where the update rule in Eq.(9) is replaced by maximizing Eq.(4)

**The implicit assumption** PALU works best when correct responses exhibit non-trivial dispersion in length. When lengths concentrate tightly (*e.g.*, when $\alpha_{0.1}$ is small for all questions), the regime update in Eq. (12) shrinks accordingly, yielding conservative (slower) reductions in $L$ while preserving performance. Empirically, we rarely observe such concentration in reasoning models (see Figure 1), though we acknowledge it as a potential limitation.

Table 2: Performance and conciseness comparison of different concise reasoning methods with DEEPSEEK-R1-DISTILL-QWEN-1.5B as the base model and DEEPSCALER as the training dataset. **P@1**: average pass@1 accuracy (%); **Tok**: average response length in tokens. **AE Score**: accuracy-efficiency score for balancing length reduction and accuracy preservation (Luo et al., 2025a).

| Model & Methods | MATH 500 | | AIME 2024 | | AIME 2025 | | AMC 2023 | | Olympiad | | Minerva-Math | | Macro Average | | AE Score ↑ |
|---|---|---|---|---|---|---|---|---|---|---|---|---|---|---|---|
| | P@1 | Tok | P@1 | Tok | P@1 | Tok | P@1 | Tok | P@1 | Tok | P@1 | Tok | P@1 | Tok | |
| R1-DISTILL-QWEN-1.5B | 82.1 | 5534 | 28.5 | 16590 | 22.3 | 16381 | 62.7 | 10615 | 43.5 | 11587 | 26.0 | 7076 | 44.2 | 11297 | 0.0 |
| *SFT- & DPO-Based* | | | | | | | | | | | | | | | |
| Kimi 1.5 SFT (Team et al., 2025a) | 68.5 | 6761 | 22.0 | 17400 | - | - | 60.4 | 9323 | 39.4 | 10036 | 23.6 | 2804 | *42.7* | *9865* | *-0.499* |
| Kimi 1.5 DPO (Team et al., 2025a) | 83.3 | 4464 | 31.7 | 13389 | - | - | 63.0 | 8678 | 44.5 | 9604 | 26.9 | 6070 | *49.9* | *8441* | *0.289* |
| TokenSkip (Xia et al., 2025) | 64.1 | 1120 | 6.8 | 2231 | - | - | 37.3 | 1401 | 25.8 | 2061 | 20.7 | 1674 | *30.9* | *1697* | *-1.173* |
| *RL-Based* | | | | | | | | | | | | | | | |
| CosFn (Yeo et al., 2025) | 75.6 | 2735 | 27.5 | 12492 | - | - | 61.1 | 6970 | 42.9 | 8307 | 27.1 | 3485 | *46.8* | *6798* | *0.249* |
| DIET (Chen et al., 2025) | 83.0 | 3061 | 31.8 | 10578 | - | - | 65.4 | 6425 | 43.7 | 6917 | 26.9 | 3505 | *50.2* | *6097* | *0.547* |
| Kimi 1.5 RL (Team et al., 2025a) | 66.3 | 1552 | 18.8 | 9109 | - | - | 44.7 | 3808 | 28.5 | 4774 | 16.7 | 1009 | *35.0* | *4050* | *-0.871* |
| O1-Pruner (Luo et al., 2025a) | 79.1 | 2531 | 25.0 | 8961 | - | - | 62.5 | 5010 | 39.0 | 5242 | 23.7 | 2400 | *45.9* | *4829* | *0.193* |
| AutoThink-Stage1 (Tu et al., 2025) | 82.1 | 2473 | 33.5 | 12716 | 22.9 | 10028 | 66.0 | 5440 | 45.6 | 7328 | 27.0 | 5372 | *46.2* +4.53% | *7226* −36.0% | 0.552 |
| AutoThink-Stage2 (Tu et al., 2025) | 85.2 | 3702 | 31.8 | 12117 | 25.6 | 11557 | 66.6 | 7415 | 46.4 | 8030 | 27.2 | 5481 | *47.1* +6.68% | *8050* −28.7% | 0.521 |
| AutoThink-Stage3 (Tu et al., 2025) | 85.1 | 1897 | 41.9 | 9033 | 25.4 | 8968 | 71.9 | 4696 | 49.0 | 5005 | 30.5 | 3834 | *50.6* +14.6% | *5563* −50.7% | 1.077 |
| ALP (Xiang et al., 2025) | 80.5 | 1435 | 37.9 | 8084 | 26.2 | 8556 | 76.5 | 3513 | 47.6 | 4670 | 24.5 | 2197 | *48.9* +10.6% | *4742* −58.0% | 0.958 |
| L1-Max (Aggarwal & Welleck, 2025) | 83.5 | 3337 | 21.7 | 4093 | 25.1 | 3792 | 66.3 | 3350 | 45.6 | 2698 | 25.2 | 2595 | *44.6* +0.80% | *3311* −70.7% | 0.551 |
| ShorterBetter (Yi et al., 2025) | 62.9 | 626 | 22.9 | 4617 | 21.0 | 4454 | 65.0 | 2311 | 34.8 | 2674 | 19.8 | 827 | *37.7* −14.6% | *2585* −77.1% | 0.030 |
| ThinkPrune-4k (Hou et al., 2025) | 83.0 | 2745 | 29.5 | 8557 | 22.6 | 7952 | 71.7 | 4241 | 45.2 | 5505 | 26.5 | 3341 | *46.4* +5.05% | *5390* −52.3% | 0.659 |
| PALU (ours) | 85.3 | 1502 | 40.0 | 7132 | 28.8 | 6869 | 81.8 | 3174 | 49.5 | 3958 | 24.2 | 1922 | **51.6** +16.8% | **4093** −63.8% | **1.185** |

## 5 EXPERIMENT

### 5.1 GENERATION LENGTH ASSUMPTION

PALU is predicated on the assumption that correct responses exhibit a *broad* distribution of lengths for given questions. If not, and the distribution were narrow, updates to the budget $L$ would converge slowly. To evaluate this key assumption, we prompt open-source reasoning models, measuring the response lengths deemed correct for a sample prompt (Figure 1). Results on more prompts, together with extended analyses for the QWEN3 and DEEPSEEK-R1 families, are reported in Figure 6 in Appendix. The observed distribution in Figure 1 reveals marked variability: the longest correct responses are two to three times longer than the shortest. This broad spread supports PALU's premise and indicates that the length budget $L$ can be progressively reduced once the performance threshold is satisfied.

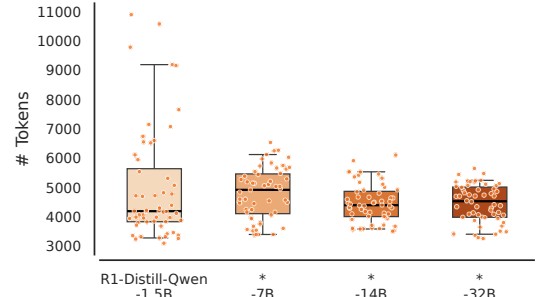

Figure 1: Token-length distribution of correct rollouts from the DEEPSEEK-R1-DISTILL-QWEN series of reasoning models. Box plots indicate the range between the 25th and 75th percentiles.

### 5.2 COMPARISON WITH EXISTING SOLUTIONS

**Model and training data.** We finetune DEEPSEEK-R1-DISTILL-QWEN-1.5B (Guo et al., 2025) and PHI-4-MINI-REASONING (Xu et al., 2025a) on a curated 12k math problems from GURU's DEEPSCALER partition, removing overly easy or difficult samples (Cheng et al., 2025; Luo et al., 2025b). **Implementation.** We implement PALU on top of VERL (Sheng et al., 2024), with the hyperparameter step size $\alpha_{0.5}$ for (rapid) length reduction and performance threshold $C = 0.8$. Training is performed for 6400 gradient update steps (roughly 1100 H200 GPU hours) for DEEPSEEK-R1-DISTILL-QWEN-1.5B and 3200 steps (850 H200 GPU hours) for PHI-4-MINI-REASONING. We apply PALU from an initial generation budget of 16k tokens (Line 4 in Algorithm 1) and update it based on Eq. (12). **Evaluation Protocol.** During the evaluation, the generation budget is set to 32k. We report Pass@1 and the output length on: MATH-500 (Hendrycks et al., 2021), AIME 24, AIME 25, AMC 2023, MINERVA, and OLYMPIADBENCH (He et al., 2024). Besides, we employ the Accuracy-Efficiency (AE) Score (Luo et al., 2025a), a composite metric balancing

Table 3: Performance and conciseness comparison on PHI-4-MINI-REASONING.

| Model & Methods | MATH 500 | | AIME 2024 | | AIME 2025 | | AMC 2023 | | Olympiad | | Minerva-Math | | Macro Average | | AE Score ↑ |
|---|---|---|---|---|---|---|---|---|---|---|---|---|---|---|---|
| | P@1 | Tok | P@1 | Tok | P@1 | Tok | P@1 | Tok | P@1 | Tok | P@1 | Tok | P@1 | Tok | |
| PHI-4-MINI-REASONING | 89.9 | 3997 | 45.6 | 13354 | 31.7 | 13733 | 84.5 | 7451 | 19.7 | 5164 | 33.5 | 5894 | 50.8 | 8265 | 0.0 |
| Overlong punishment (Yu et al., 2025) | 88.1 | 2795 | 28.5 | 3824 | 22.3 | 4019 | 79.7 | 5524 | 43.5 | 6065 | 30.0 | 4954 | 48.6 | 4539 | -0.15 |
| ThinkPrune-4k (Hou et al., 2025) | 88.9 | 2819 | 35.5 | 6163 | 29.2 | 5941 | 82.5 | 4764 | 42.2 | 5135 | 31.3 | 2677 | 51.6 | 4583 | -0.01 |
| PALU (Ours) | 89.3 | 1929 | 45.3 | 7423 | 30.7 | 7152 | 84.6 | 3241 | 49.8 | 3900 | 34.3 | 2056 | 55.7 | 4283 | 0.27 |

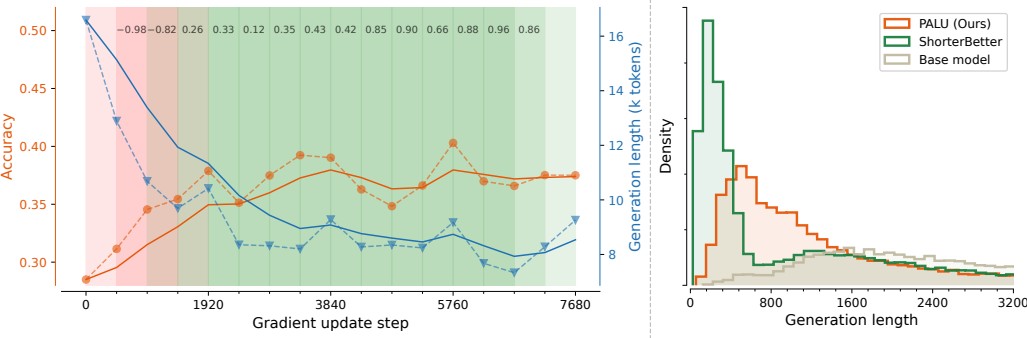

Figure 2: *Left:* Performance-conciseness evolution of PALU. The evaluation dataset is AIME24. We encode their *Spearman's* correlations with red (negative) and green (positive) regions. *Right:* Distribution of generation lengths under PALU and ShorterBetter (Yi et al., 2025).

length reduction against accuracy preservation, for overall comparison. We report results averaged over 32 rollouts for AIME 24, AIME 25, and 10 for others, following Chen et al. (2025).

**Comparison results on DEEPSEEK-R1-DISTILL-QWEN-1.5B (Table 2)** We consider two families of baselines. *(i) SFT/DPO-based models*: Kimi k1.5 SFT, Kimi k1.5 DPO (Team et al., 2025a), and TokenSkip (Xia et al., 2025). *(ii) RL-based methods*: reward-function-based methods such as CosFN (Yeo et al., 2025), Kimi k1.5 RL (Team et al., 2025a), DIET (Chen et al., 2025), ShorterBetter (Yi et al., 2025), L1-Max (Aggarwal & Welleck, 2025), and ALP (Xiang et al., 2025); stage-based length budgeting methods that progress shrink the rollout budget, for example, ThinkPrune (Hou et al., 2025); and multi-stage RL pipelines, *e.g.,* AutoThink (Tu et al., 2025).

*PALU achieves superiority in both conciseness and accuracy (Table 2).* Across six math and science tasks, PALU reduces the macro-average response length from 11,297 to 4093 tokens, a 63% reduction. Meanwhile, it surpasses other RL-based methods in terms of accuracy. The consistent gains in accuracy and conciseness highlight the effectiveness of the *Lagrangian* optimization objective.

*PALU reduces both easy and hard redundancies (Figure 2, left).* We monitor the joint evolution of evaluation accuracy and generation length throughout training, with *Spearman* correlations between the two encoded by color (window size 4). In the initial phase (red-shaded), accuracy rises as length falls, showing that PALU eliminates redundant tokens without harming performance. As training progresses, the correlation turns positive (green-shaded), revealing a genuine trade-off: further compression now risks eroding accuracy. This marks the *harder redundancies*. PALU responds adaptively, retaining moderately longer responses when beneficial while continuing to shorten those that can be solved concisely. Consequently, the overall generation length continues to decline (solid curves), even under trade-off pressure. These dynamics demonstrate that PALU not only captures the low-hanging fruit of trivial redundancy removal but also sustains balanced improvements in the more challenging regime where performance and conciseness must be carefully reconciled.

*PALU retains moderate-length responses when beneficial (Figure 2, right).* We then present the generation length distributions of PALU and ShorterBetter (Yi et al., 2025) on evaluations tasks in Figure 2. ShorterBetter, as a reward-based method, produces a sharp peak at very short lengths (less than 320 tokens) and very few responses in the middle range around 800 tokens, suggesting it often cuts too aggressively. In contrast, PALU spreads its density more evenly, keeping many responses in the moderate range while still limiting very long outputs. This pattern reflects PALU's strength: it avoids excessive shortening while still trimming unnecessary length, which helps preserve accuracy.

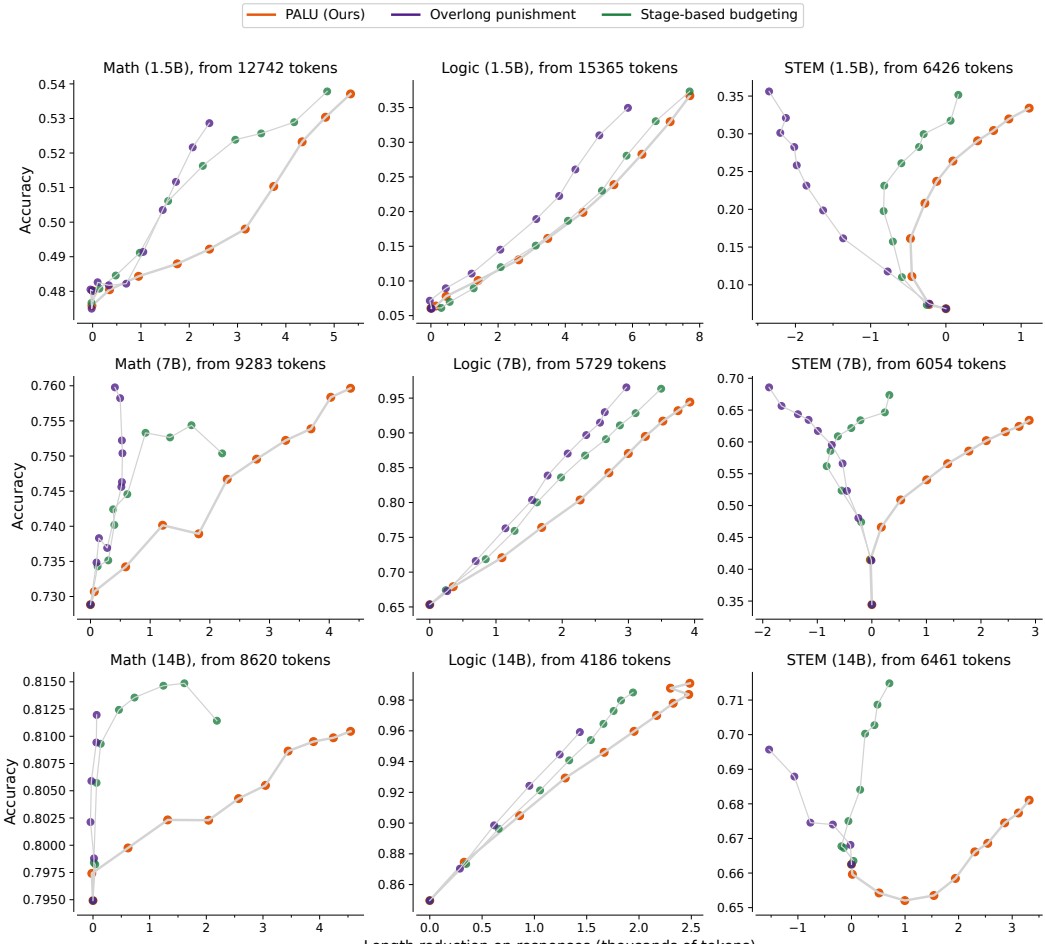

Figure 3: Conciseness-performance evolution of DEEPSEEK-R1-DISTILL-QWEN-1.5B trained with different concise reasoning methods. The training dataset covers three-domain questions: math, logic and STEM. Results are plotted with time weight exponential moving average smoothing.

**Comparison results on PHI-4-MINI-REASONING (Table 3)** To assess the generality of PALU across model architectures, we further evaluate it on PHI-4-MINI-REASONING. Owing to the computational cost, we include two representative baselines: ThinkPrune-4k, and overlong-punishment (Yu et al., 2025), which penalizes responses exceeding 4k tokens. All models are trained with a context length of 8k, and the results are summarized in Table 3. PALU attains comparable accuracy to the base model while maintaining a shorter generation length, in contrast to alternative methods that sacrifice accuracy for brevity. This superior balance between accuracy and conciseness on PHI-4-MINI-REASONING confirms PALU's robust and architecture-agnostic effectiveness.

5.3 SCALING TO MULTI-DOMAIN TASKS AND LARGE MODELS

**Multi-domain and multi-scale comparison (Figure 3)** To examine PALU's adaptivity on domains and model scales, we conduct comparison using a series of DEEPSEEK-R1-DISTILL-QWEN models with parameters 1.5B, 7B, and 14B, with the training data covering math, logic and STEM from the GURU(Cheng et al., 2025) dataset. We limit the training data to 5, 120 samples (2k math, 2k STEM and 1k logic) and train the model for only 10 epochs. For evaluation, we use another 768 questions spanning math, logic, and STEM, and report both accuracy (pass@1 over 10 rollouts) and generation length reductions (in thousands of tokens) on test partitions. For comparison, we employ *(i)* stage-based budgeting from Hou et al. (2025) with gradually reducing the generation-length budget from 16k to 8k over five stages; and *(ii)* soft overlong punishment strategy introduced by DAPO (Yu et al., 2025), with a penalty for responses with length exceeding 8k. These approaches serve as representatives of length-budget-based and reward-function-based methods.

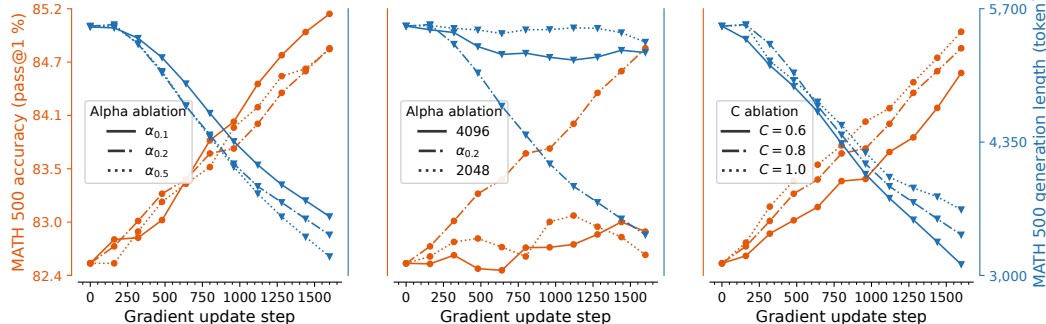

Figure 4: Ablation study on the step size $\alpha_\tau$ (left, middle) and the performance threshold $C$ (right).

*PALU adapts across data domains and model scales (Figure 3).* All three methods improve accuracy on the in-distribution test sets. Yet, their impact on conciseness diverges. **The multi-domain scenario.** Consider the 1.5B model (first row of Figure 3). Stage-based budgeting and overlong punishment shorten responses for math and logic tasks, with evaluation curves showing clear progress to the right-hand side (*i.e.*, gains in length reduction). Yet in STEM, these heuristics fail. Their reliance on a fixed target length (8k in our implementation) leaves little space for further reduction, as the base model already generates shorter responses ($\sim$ 6.5k tokens), well below the assumed optimum. **The multi-scale scenario.** Initial generation length varies substantially across model sizes, especially for math and logic tasks (as indicated in subtitles for the left column of Figure 3). This variation poses a fundamental challenge for heuristic methods: because they require an explicit length target, each new model scale demands repeated trial-and-error sweeps to locate a workable setting. **PALU.** Rather than imposing heuristic length targets, PALU dynamically adjusts its budget under a joint conciseness-performance objective. This principled formulation, grounded in *Lagrangian* dynamics, adapts seamlessly to varying initial length distributions and performance-length trade-offs. As a result, PALU achieves consistent improvements across domains and model sizes. In short, heuristic approaches work in narrow cases but break down when domain or model characteristics shift. PALU avoids this brittleness by treating concise reasoning as a performance-constrained optimization problem, delivering robust conciseness and accuracy gains across diverse settings.

### 5.4 ABLATION AND COMPUTATIONAL OVERHEAD ANALYSIS

PALU draws inspirations from the *Lagrangian* formulation for minimizing the length budget for rollouts. This provides a principled yet efficient budgeting mechanism involving two hyperparameters: (1) a step size $\alpha_\tau$, and (2) a performance threshold $C$:

$$L = \begin{cases} L - \alpha_\tau^{(q)} & \text{if } R \geq C \\ L_{\max} & \text{otherwise} \end{cases}.$$

**Step size in PALU (Figure 4, left)** $\alpha_\tau^q$ measures the gap between the longest correct response and the $(1 - \tau)$-quantile length for the question $q$. $\tau$ directly determines the step size for updating $L$. To examine its sensitivity and guide practitioners, we conduct an ablation study across different $\tau$ values with fixed performance threshold $C = 0.8$. Using the multi-domain dataset (math, logic, and STEM), we run PALU with update steps $\alpha_{0.1}$, $\alpha_{0.2}$, and $\alpha_{0.5}$. We evaluate the model on the MATH 500 benchmark and report the accuracy and generation length during the training process. As shown in Figure 4, left, a larger step ($\alpha_{0.5}$) accelerates length reduction but slightly compromises accuracy, whereas smaller steps stabilize performance but provide weaker pressure for conciseness.

**Replacing $\alpha_\tau$ with heuristic values (Figure 4, middle)** The *Lagrangian*-based formulation in PALU provides a principled mechanism for adjusting the length budget. To highlight its advantage, we replace the step size $\alpha_\tau$ with fixed values, setting $\Delta L \in \{2048, 4096\}$. In this setup, once the model meets the performance threshold (*e.g.*, $C = 0.8$), the length budget is reduced by a fixed amount; otherwise, it is reset to the maximum budget (16k). The comparison results in the middle panel of Figure 4 show that PALU's adaptive step size allows the model to shorten outputs smoothly and stably, while the heuristic fixed-step variants struggle to balance conciseness and performance. This confirms the importance of PALU's principled update rule for stable optimization.

**Performance threshold in PALU (Figure 4, right)**  The threshold $C$ serves as a constraint boundary between samples that should receive more or less generation budget and thus balances accuracy and conciseness. We validate this intuition by sweeping $C \in \{0.6, 0.8, 1.0\}$ and reporting results on the MATH-500 task in the right panel of Figure 4. A high threshold *e.g.,* $C = 1.0$, drives the optimization toward performance recovery, preserving accuracy at the cost of longer outputs, while a lower threshold, $C = 0.6$, yields the most compact generations with a modest drop in accuracy. The intermediate setting, $C = 0.8$, achieves the balance between the two objectives.

**Computational overhead analysis (Figure 5)**
On the memory side, PALU introduces three arrays and one scalar to record: (1) pass rates from the previous epoch, (2) the interested length quantiles of correct responses in the previous epoch, (3) length budgets for each prompt, and (4) a global performance threshold. This requires about 144 KB for a dataset of 12 thousand of prompts. On the computation side, PALU assigns each prompt a generation length (budget) that progressively drives towards concise reasoning traces compared to the GRPO baseline. Besides, since the length budget is assigned before conducting the rollouts, it is therefore to sort the prompts by their budget to form balanced batches. Together, these two factors reduce idle time and computation during inference, thereby improving overall efficiency by approximately 9.2% compared to the GRPO implementation, as shown in Figure 5.

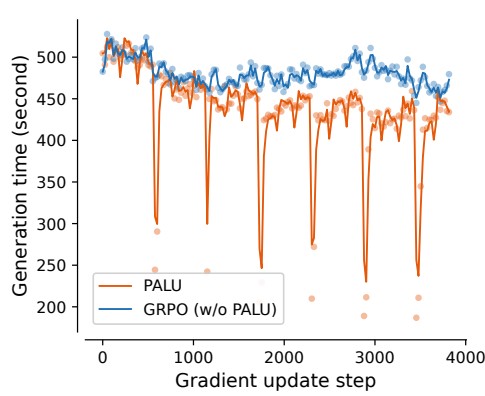

Figure 5: Rollout time comparison (training).

## 6 LIMITATIONS AND CONCLUSION

**Limitation**  PALU assumes that overthinking LLMs have a broad distribution of response lengths. While we empirically verified this property in our experiments, we acknowledge an extreme case where the model always generates responses of identical length. In such a scenario, even reducing the length budget by a single token could cause accuracy to collapse from 1.0 to 0.0, rendering PALU ineffective. Another limitation is that we do not claim PALU's concise reasoning behavior will generalize to out-of-distribution domains. We view such generalization as stemming primarily from the diversity of training data and the RL component, rather than from PALU itself.

**Conclusion**  Although these limitations define the scope of our study, they do not detract from our central contribution: a principled and pragmatic solution for concise reasoning. Although trimming overly long responses seems intuitive, achieving this without compromising accuracy and while retaining adaptivity across domains and model scales calls for a principled formulation. PALU elevates the intuition into theory by casting the task as a constrained optimization and resolving it through the *Lagrangian* framework. This shift from intuitive observation to principled methodology constitutes PALU's broader contribution to the community. Technically, it affords two advantages. First, PALU automatically balances conciseness and performance without ad-hoc heuristics, reducing generation length by 64% while improving accuracy by 16% across six benchmark tasks. Second, it provides a principled update rule for the length budget, enabling robust adaptation across domains (math, logic, STEM) and model scales (1.5B, 7B, and 14B parameters).

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

## A   CONFIGURATIONS FOR TRAINING AND EVALUATION

**Training recipe**   We integrate our PALU strategy to the VeRL implementation of GRPO and fine-tune DEEPSEEK-R1-DISTILL-QWEN-1.5B, 7B, and 14B models using the following recipe:

Table 4: Training recipe for finetuning DEEPSEEK-R1-DISTILL-QWEN-1.5B, 7B, and 14B.

| Parameter | Value |
|---|---|
| Learning rate | $1e-6$ |
| Rollout batch size (prompts) | 512 |
| Gradient update batch size (prompts) | 32 |
| KL-divergence coefficient | 0.0 |
| Max response length | 16k |
| Loss aggregation mode | token-loss |
| Clip ratio low | 0.2 |
| Clip ratio high | 0.28 |
| Number of rollouts per sample | 8 |
| *Length update step size (Table 2, Figure 2) | $\alpha_{0.5}$ |
| *Length update step size (Figure 3) | $\alpha_{0.2}$ |
| *Performance threshold C | 0.8 |

**Training datasets**   For training, we employ two types of datasets:

- 12k mathematics question-answer pairs for the run in **Table 2, Figure 2, and Table 7** (benchmarking comparison and its in-depth analysis). This dataset is a slice from the GURU's DEEPSCALER partition. We train DEEPSEEK-R1-DISTILL-QWEN-1.5B for 20 epochs on it. This dataset is used to compare performance.

- 5k multi-domain questions for the comparisons in **Figure 3**, **Figure 4** and **Figure 5**. We randomly select *(i)* 2k math samples from the DEEPSCALER partition, *(ii)* 2k STEM samples from the STEM-web partition and *(iii)* 1k logic questions from the logic ordering puzzle partition of the GURU collection. We train DEEPSEEK-R1-DISTILL-QWEN-1.5B, 7B, and 14B for 10 epochs for the multi-domain comparison and the ablation study. This dataset is used to analyze training dynamics.

**Compute resources**   We conduct our experiments on H200 GPUs clusters. Results in Table 2 are from DEEPSEEK-R1-DISTILL-QWEN-1.5B trained on 12k DEEPSCALER questions, which takes 2 nodes (16 GPUs) for 1100 GPU hours. Results in Figure 3 are from DEEPSEEK-R1-DISTILL-QWEN-1.5B, 7B, and 14B models trained on 5k multi-domain questions, which takes 2 nodes, 4 nodes and 8 nodes for roughly 300, 700, and 2300 GPU hours.

**Evaluation protocol**   We follow the standard decoding protocol used in concise reasoning research as listed in Table 5. For the rollout numbers, we collect 32 responses and report their statistics for the small dataset (AIME24) and 10 responses for others.

Table 5: Decoding parameters.

| Parameter | Value |
|---|---|
| Temperature | 0.6 |
| Top_p | 0.95 |
| Top_k | - |
| Max response length | 32k |

Table 6: Number of Rollouts for reporting the averaged performance and generation length.

| Dataset partition | Number of rollouts (for evaluation) |
|---|---|
| AIME 24 | 32 |
| Others (MATH 500, AMC 23, *etc.*) | 10 |

Table 7: Detailed Accuracy-Efficiency (AE) Score comparison.

| Methods/Model | MATH 500 | AIME24 | AIME 2025 | AMC23 | Olympiad | MinervaMath | Marco Average ↑ |
|---|---|---|---|---|---|---|---|
| R1-Distill-Qwen-1.5B | 0.000 | 0.000 | 0.000 | 0.000 | 0.000 | 0.000 | 0.00 |
| Kimi 1.5 SFT | -1.050 | -1.189 | - | -0.062 | -0.337 | 0.142 | *-0.499* |
| Kimi 1.5 DPO | 0.237 | 0.530 | - | 0.197 | 0.240 | 0.246 | *0.290* |
| TokenSkip | -0.299 | -2.941 | - | -1.158 | -1.212 | -0.256 | *-1.173* |
| CosFn | 0.110 | 0.072 | - | 0.216 | 0.214 | 0.634 | *0.249* |
| DIET | 0.480 | 0.710 | - | 0.524 | 0.417 | 0.609 | *0.548* |
| Kimi 1.5 RL | -0.243 | -1.251 | - | -0.794 | -1.136 | -0.931 | *-0.871* |
| O1-Pruner | 0.360 | -0.154 | - | 0.512 | 0.030 | 0.219 | *0.193* |
| GRPO | 0.102 | 1.124 | 0.923 | 1.136 | 0.590 | -0.496 | 0.563 |
| AutoThink-Stage1 | 0.553 | 0.760 | 0.470 | 0.645 | 0.512 | 0.356 | 0.552 |
| AutoThink-Stage2 | 0.444 | 0.617 | 0.740 | 0.488 | 0.507 | 0.364 | 0.521 |
| AutoThink-Stage3 | 0.767 | **1.866** | 0.870 | 0.998 | 0.947 | **0.977** | 1.077 |
| ALP | 0.643 | 1.502 | 1.000 | 1.329 | 0.880 | 0.401 | 0.958 |
| L1-Max | 0.448 | -0.440 | 1.150 | 0.857 | 0.912 | 0.479 | 0.551 |
| ShorterBetter | -0.282 | -0.261 | 0.440 | 0.892 | -0.231 | -0.309 | 0.030 |
| ThinkPrune | 0.536 | 0.589 | 0.550 | 1.031 | 0.642 | 0.585 | 0.659 |
| PALU (ours) | **0.846** | 1.781 | 1.460 | **1.615** | **1.072** | 0.382 | **1.185** |

**Accuracy-Efficiency (AE) Score (in Table 2)** To evaluate whether a model improves inference efficiency, in other words, producing shorter responses without sacrificing accuracy, we adopt the *Accuracy-Efficiency (AE) Score*, introduced by Luo et al. (2025a). This metric combines the length reduction in response length and the accuracy improvement into a single number. It is formally defined as

$$\text{AE Score} = \begin{cases} \varphi \cdot \Delta\text{Length} + \eta \cdot |\Delta\text{Acc}|, & \text{if } \Delta\text{Acc} \geq 0 \\ \varphi \cdot \Delta\text{Length} - \theta \cdot |\Delta\text{Acc}|, & \text{if } \Delta\text{Acc} < 0 \end{cases},$$

where the terms are defined as follows:

- **Length reduction ratio:**

$$\Delta\text{Length} = \frac{\text{Length}_{\text{base}} - \text{Length}_{\text{model}}}{\text{Length}_{\text{base}}}.$$

  A positive $\Delta\text{Length}$ indicates the evaluated model produces shorter outputs than the base model.

- **Accuracy change ratio:**

$$\Delta\text{Acc} = \frac{\text{Acc}_{\text{model}} - \text{Acc}_{\text{base}}}{\text{Acc}_{\text{base}}}.$$

  $|\Delta\text{Acc}|$ measures the relative magnitude of accuracy gain or drop against the base model.

Positive AE Scores reflect desirable improvements: generating shorter outputs while maintaining or improving accuracy. Negative AE Scores arise when accuracy degradation outweighs the benefit of shorter responses. We follow Luo et al. (2025a) and adopt the same hyperparameters:

- $\varphi = 1$ (weight on length reduction),
- $\eta = 3$ (bonus for accuracy gains),
- $\theta = 5$ (penalty for accuracy drops).

The asymmetric weighting ($\theta > \eta$) ensures that accuracy drops are penalized more heavily than accuracy gains are rewarded, aligning with the practical preference to avoid performance degradation even when outputs become shorter.

We provide the detailed comparison of AE Score in Table 7 for reference.

## B SOME EMPIRICAL EVIDENCE

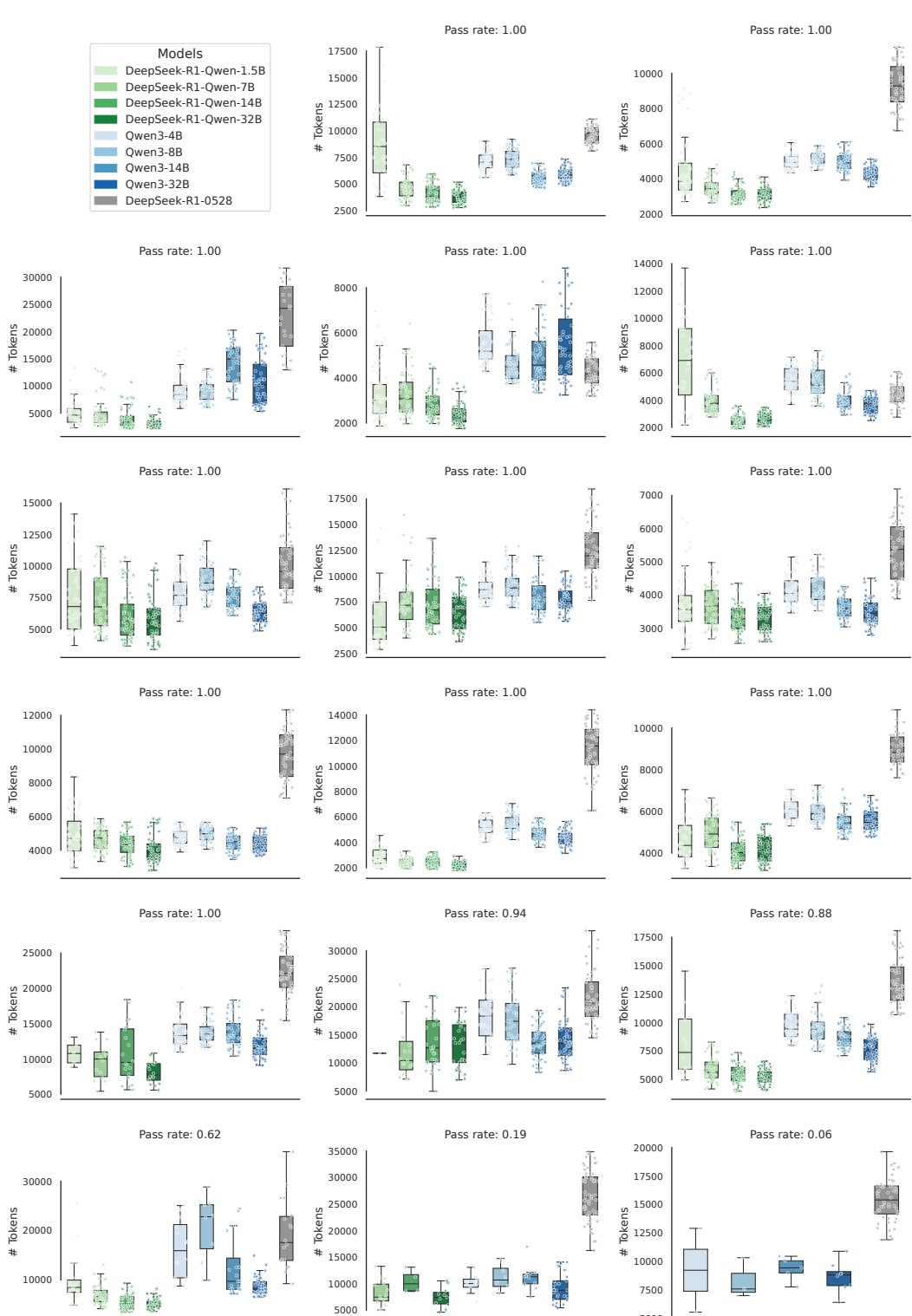

Figure 6: **Overthinking LLMs exhibit broad variation in the length of (correct) generations (Figure 1).** Token-length distributions of correct responses from open-source reasoning LLMs (DEEPSEEK-R1-DISTILL-QWEN, QWEN3, and DEEPSEEK-R1-0528) on randomly selected 18 questions from the GURU dataset. Box plots show the interquartile range (25th–75th percentiles).

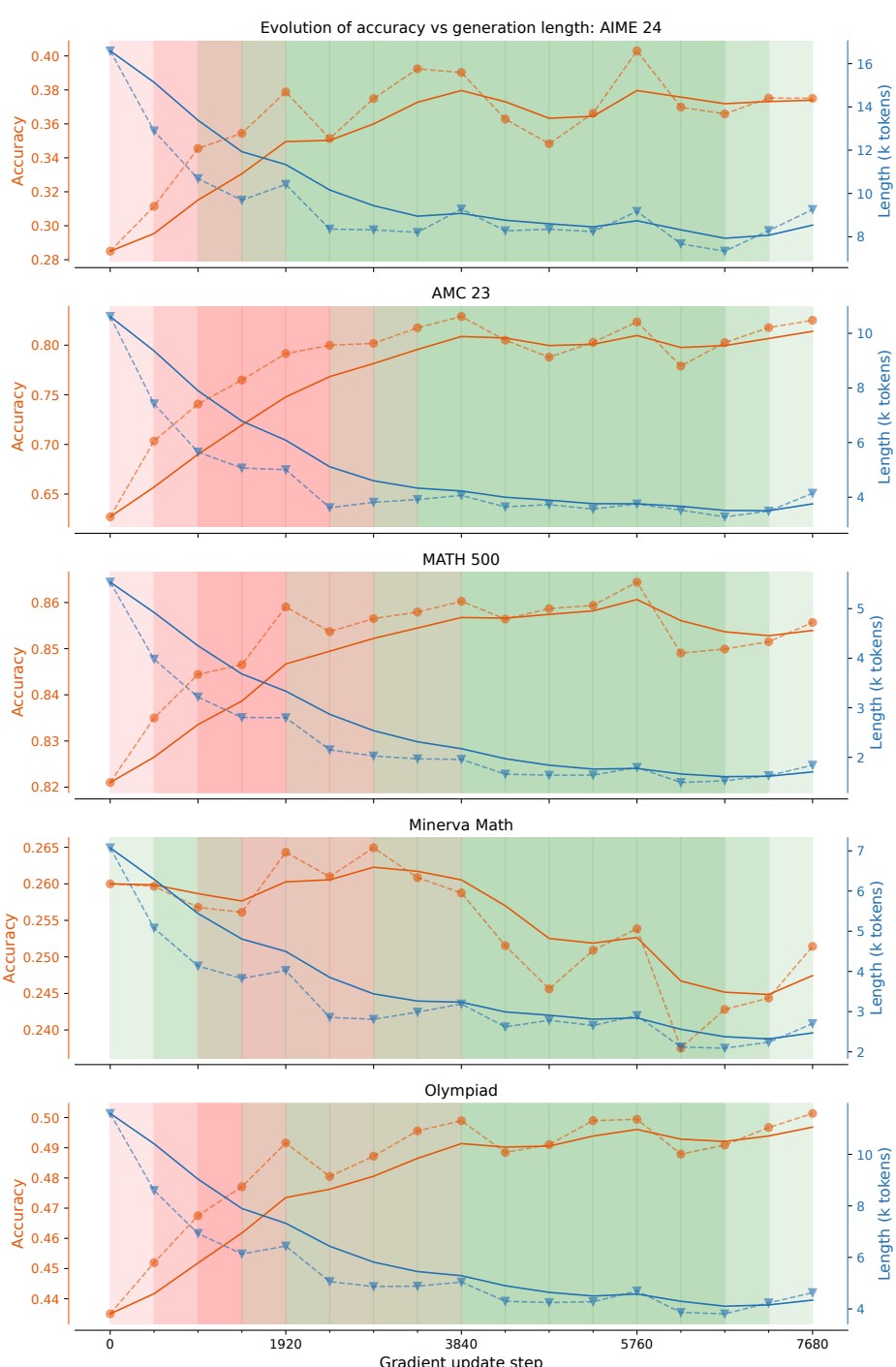

Figure 7: **PALU reduces both easy and hard redundancies (Figure 2).** Performance–conciseness evolution during PALU's training. We encode the *Spearman* correlation between performance and generation length using red (negative) and green (positive) colors. In the early phase, the two are negatively correlated: accuracy rises while length decreases. As training progresses, the correlation becomes positive, indicating further shortening begins to limit accuracy. Nevertheless, PALU continues to reduce generation length even in this harder regime, as shown by the overall solid curves.

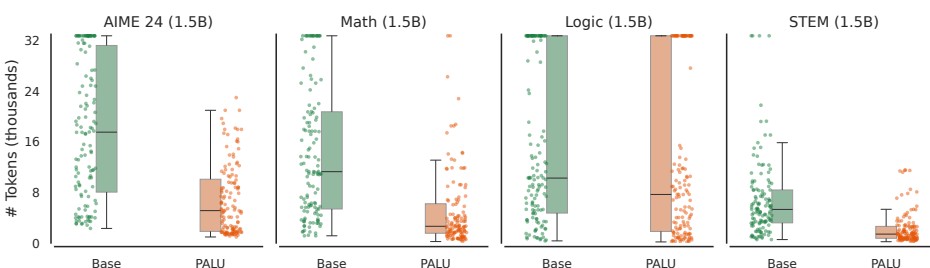

Figure 8: **Different question domains exhibit distinct generation length distributions.** We plot the length distributions of responses from the base model (DEEPSEEK-R1-DISTILL-QWEN-1.5B) and the model finetuned with PALU (on math data, specifically, the 12k DEEPSCALER subset). Scatter points show raw lengths, while boxplots indicate the interquartile range (25th–75th percentiles). The base model produces shorter responses on STEM-domain questions but much longer ones on AIME 24 questions.

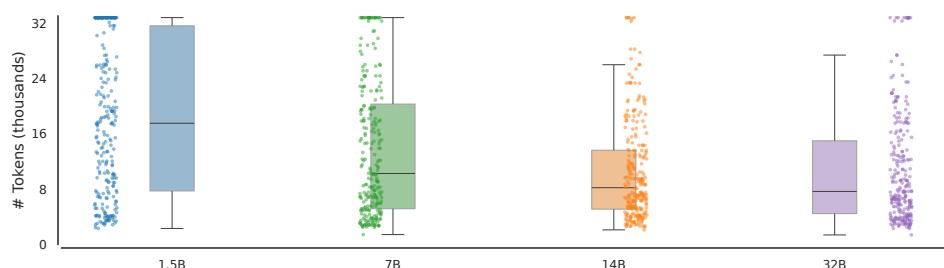

Figure 9: **Different model scales exhibit distinct generation length distributions.** We plot the response length distributions of models from the DEEPSEEK-R1-DISTILL-QWEN series with parameter 1.5B, 7B, 14B and 32B. Scatter points show raw lengths, while boxplots indicate the interquartile range (25th–75th percentiles). The evaluation dataset is AIME 2024.

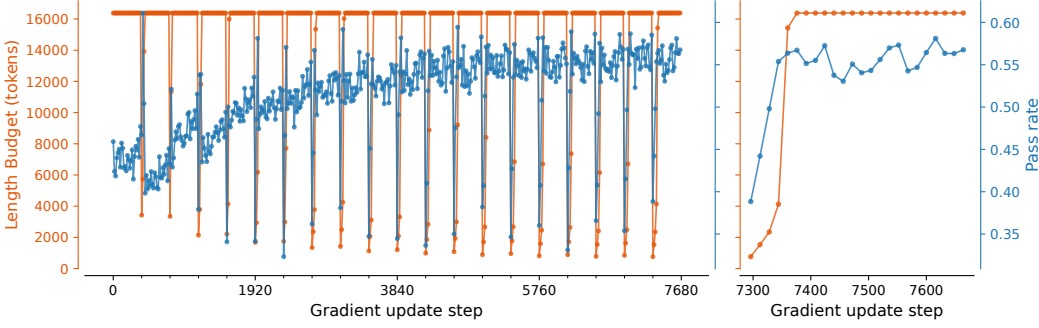

Figure 10: **Evolution of length budget (orange) and on-policy pass rate (blue) during training.** The model is DEEPSEEK-R1-DISTILL-QWEN-1.5B, trained on a 12k-sample dataset. The right panel zooms in on the final training epoch for better visualization. Throughout training, the on-policy pass rate shows a steady upward trend. In the last epoch, among the 24 generation batches, five exhibit a reduced length budget, while the rest maintain the maximum 16k-token budget.

## C    GENERATION EXAMPLES

**Question 1: Which one is greater, 8.11 or 8.9?**

**DeepSeek-R1-Distill-Qwen-1.5B:**

*First, I need to compare the whole number parts of both numbers. Both 8.11 and 8.9 have the same whole number part, which is 8. Since the whole numbers are equal, I'll move on to comparing the decimal parts. Looking at 8.11, the decimal part is 0.11, while the decimal part of 8.9 is 0.9. To make the comparison easier, I can rewrite 0.9 as 0.90. Now, comparing 0.11 and 0.90, it's clear that 0.90 is larger. Therefore, 8.9 is greater than 8.11.*

*Solution:*

*To determine which number is larger between 8.11 and 8.9, follow these steps:*

*1. Compare the Whole Numbers: - Both numbers have the same whole number part: 8.*

*2. Compare the Decimal Parts: - Convert 8.9 to have two decimal places: 8.90. - Now, compare 0.11 (from 8.11) and 0.90 (from 8.90).*

*3. Determine Which Decimal is Larger: - 0.90 is greater than 0.11.*

*4. Conclusion: - Since 0.90 > 0.11, 8.90 is greater than 8.11.*

*Therefore, 8.9 is larger than 8.11.*

$\boxed{8.9}$

**PALU:**

*To determine which number is larger between 8.11 and 8.9, I'll compare their decimal places. Both numbers have the same whole number part, which is 8. Next, I'll look at the tenths place. In 8.11, the tenths digit is 1, while in 8.9, it's 9. Since 9 is greater than 1, 8.9 is larger than 8.11.*

*To determine which number is larger between 8.11 and 8.9:*

*1. Compare the Whole Number Part: - Both numbers have the same whole number part, which is 8.*

*2. Compare the Tenths Place: - 8.11 has a tenths digit of 1. - 8.9 has a tenths digit of 9. Since 9 is greater than 1, 8.9 is larger than 8.11.*

$\boxed{8.9}$

**Question 2: What is the first prime number after two hundred?**

**DeepSeek-R1-Distill-Qwen-1.5B:**

*To determine the first prime number after 200, I'll start by checking each number sequentially to see if it's prime. I'll begin with 201 and check if it's divisible by any prime numbers less than its square root. The square root of 201 is approximately 14.17, so I only need to check divisibility by primes up to 13. Next, I'll check 202. Since it's even, it's divisible by 2, so it's not prime. Then, 203: I'll check if it's divisible by 7, 11, and 13. 203 divided by 7 is 29, so it's not prime. After that, 204 is even, so it's divisible by 2. 205 ends with a 5, so it's divisible by 5. 206 is even, so it's divisible by 2. 207: Adding the digits gives 9, which is divisible by 3, so 207 is not prime. 208 is even, so it's divisible by 2. 209: Checking divisibility by 11, since 209 divided by 11 is 19, it's not prime. Finally, 210 is even, so it's divisible by 2. 211: Checking divisibility by primes up to 14, it doesn't divide evenly by 2, 3, 5, 7, 11, or 13. Therefore, 211 is the next prime number after 200.*

*To find the first prime number after 200, follow these steps:*

*1. Understand What a Prime Number Is: - A prime number is a natural number greater than 1 that has no positive divisors other than 1 and itself.*

*2. Start Checking Numbers After 200: - Begin with the number 201 and check each subsequent number to see if it's prime.*

*3. Check Divisibility: - 201: - 201 / 3 = 67 - Since 201 is divisible by 3, it's not prime. - 202: - It's even, so divisible by 2. - Not prime. - 203: - Check divisibility by primes up to sqrt203 = 14.25. - 203 / 7 = 29, so it's not prime. - 204: - It's even. - Not prime. - 205: - Ends with 5, divisible by 5. - Not prime. - 206: - It's even. - Not prime. - 207: - Sum of digits is 9, divisible by 3. - Not prime. - 208: - It's even. - Not prime. - 209: - Check divisibility by primes up to sqrt209 = 14.45. - 209 / 11 = 19, so it's not prime. - 210: - It's even. - Not prime. - 211: - Check divisibility by primes up to sqrt211 = 14.52. - 211 is not divisible by 2, 3, 5, 7, 11, or 13. - Conclusion: 211 is a prime number.*

*4. Final Answer: The first prime number after 200 is:* $\boxed{211}$

**PALU:**

*To determine the first prime number after two hundred, I'll start checking each number sequentially from 201 onwards.201 is divisible by 3, so it's not prime.202 is even, thus not prime.203 is divisible by 7, so it's not prime.204 is even, not prime.205 ends with a 5, so it's divisible by 5, not prime.206 is even, not prime.207 is divisible by 3, not prime.208 is even, not prime.209 is divisible by 11, so it's not prime.210 is even, not prime.211 is a prime number because it isn't divisible by any number other than 1 and itself. Therefore, the first prime number after two hundred is 211.*

*The first prime number after 200 is* $\boxed{211}$ *.*

**Question 3 (the first AIME 2025 problem): Find the sum of all integer bases $b > 9$ for which $17_b$ is a divisor of $97_b$.**

**DeepSeek-R1-Distill-Qwen-1.5B:**

*Okay, so I have this problem where I need to find the sum of all integer bases $b > 9$ for which $17_b$ is a divisor of $97_b$. Hmm, let me think about how to approach this. First, I know that numbers in base $b$ can be converted to decimal (base 10) by expanding each digit multiplied by powers of $b$. So, $17_b$ should be equal to $1 * b + 7 * 1$, which is $b + 7$. Similarly, $97_b$ would be $9 * b + 7x1$, so that's $9b + 7$. The problem says that $17_b$ divides $97_b$. In decimal terms, this means that $b + 7$ is a divisor of $9b + 7$. So, mathematically, we can write this as: $\frac{9b+7}{b+7}$ is an integer Let me denote this fraction as $k$, where $k$ is an integer. So, $k = \frac{9b+7}{b+7}$ To find $k$, I can perform polynomial division or rearrange the equation. Let me try rearranging it: Multiply both sides by $b + 7$ : $9b + 7 = k(b + 7)$ Expanding the right side: $9b + 7 = kb + 7k$ Now, let's bring all terms to one side: $9b + 7 - kb - 7k = 0$ Factor out $b$ from the terms that contain it: $b(9 - k) + (7 - 7k) = 0$ Let me solve for $b$ : $b(9 - k) = 7k - 7$ So, $b = \frac{7k-7}{9-k}$ Simplify numerator: $7(k - 1)$ So, $b = \frac{7(k-1)}{9-k}$. Since $b$ must be an integer greater than 9, the right side must evaluate to an integer. Also, $b > 9$ , so we need to find integer values of $k$ such that $b$ is an integer greater than 9. Let me denote $m = k - 1$, so $k = m + 1$. Substitute back into the equation: $b = \frac{7m}{9-(m+1)} = \frac{7m}{8-m}$ So now, $b = \frac{7m}{8-m}$. Since $b$ must be a positive integer greater than 9, both numerator and denominator must be integers, and $8 - m$ must divide $7m$. So, $8 - m$ divides $7m$ . Let me write this as: $8 - m \mid 7m$ Which implies that $8 - m$ divides $7m$ . Since $8 - m$ is a divisor, let me denote $d = 8 - m$ . Then, $d$ divides $7m$, which is $7(8 - d)$ . So, $d \mid 7(8 - d)$ Which is: $d \mid 56 - 7d$. Therefore, $d \mid 56 - 7d + 7d$ (since $d \mid 7d$, so adding it doesn't change divisibility). Simplifying: $d \mid 56$ So, $d$ is a positive integer divisor of 56. Since $d = 8 - m$, and $m$ is an integer (because $k$ is an integer, so $m = k - 1$ is also an integer), we can find possible values of $d$.Let me list all positive divisors of 56:1, 2, 4, 7, 8, 14, 28, 56.So, $d \in \{1, 2, 4, 7, 8, 14, 28, 56\}$.But remember that $d = 8 - m$, and $m = k - 1$. Also, since $k$ is positive (as it's a divisor in $9b + 7 = k(b + 7)$), $m$ must be positive or zero, but since $d = 8 - m$, and $d$ is positive, we have $8 - m > 0$, so $m < 8$. Therefore, $m$ can be 0, 1, 2, 3, 4, 5, 6, or 7.So, let's find corresponding $d$ for each $m$:1. $m = 0$: $d = 8 - 0 = 82$. $m = 1$: $d = 8 - 1 = 73$. $m = 2$: $d = 8 - 2 = 64$. $m = 3$: $d = 8 - 3 = 55$. $m = 4$: $d = 8 - 4 = 46$. $m = 5$: $d = 8 - 5 = 37$. $m = 6$: $d = 8 - 6 = 28$. $m = 7$: $d = 8 - 7 = 1$But wait, earlier we concluded that $d$ must be a positive divisor of 56, which is the case here since all... (exceed 32k )*

**PALU:**

*Okay, so I need to find the sum of all integer bases $b > 9$ where 17 base $b$ divides 97 base $b$. Hmm, let me break this down.First, let's convert both numbers from base $b$ to decimal.17 base $b$ is equal to $1 * b + 7$. Similarly, 97 base $b$ is $9 * b + 7$.So we need $17_b$ divides $97_b$. Translating to decimal:$1 * b + 7$ divides $9 * b + 7$.So the condition is $(b + 7)$ divides $(9b + 7)$.So let's express this as $(9b + 7)$ divided by $(b + 7)$ must be an integer.Let me compute the division:Divide $9b + 7$ by $b + 7$.Using polynomial division:Divide $9b + 7$ by $b + 7$.So $9b$ divided by $b$ is 9. Multiply $(b + 7)$ by 9: $9b + 63$.Subtract this from $9b + 7$: $(9b + 7) - (9b + 63) = 7 - 63 = -56$.So the division gives 9 with a remainder of -56.Therefore, $(9b + 7) = 9 * (b + 7) - 56$.So for $(b + 7)$ to divide $9b + 7$, $(b + 7)$ must divide the remainder, which is -56.But since divisibility is concerned with absolute value, so $(b+7)$ divides 56.So $b+7$ is a positive divisor of 56.Given that $b > 9$, so $b+7 > 16$.So $b+7$ is a divisor of 56 greater than 16.Let's list the divisors of 56:1, 2, 4, 7, 8, 14, 28, 56.Which of these are greater than 16? 28 and 56.So possible $b + 7 = 28$ or 56.Therefore $b = 21$ or 49.So the bases are 21 and 49.Sum is $21 + 49 = 70$.Hence the answer is 70.**Final Answer**The sum of all such bases is $\boxed{70}$.*

*To find the sum of all integer bases $b > 9$ for which $17_b$ is a divisor of $97_b$, we convert these numbers from base $b$ to decimal: $17_b = 1 \cdot b + 7 = b + 7$- $97_b = 9 \cdot b + 7$. We need $b + 7$ to divide $9b + 7$. Using polynomial division, we find that the remainder when $9b + 7$ is divided by $b + 7$ is -56. Thus, $b + 7$ must be a divisor of 56.Given $b > 9$, we find the divisors of 56 greater than 16: 28 and 56. Therefore, $b + 7 = 28$ or $b + 7 = 56$, giving $b = 21$ or $b = 49$.The sum of these bases is $21 + 49 = 70$.Thus, the sum of all such bases is $\boxed{70}$.*

## D    THE USE OF LARGE LANGUAGE MODELS

An LLM was used to refine writing for clarity and readability but did not contribute to research design, experiment, or analysis. All intellectual work was independently conducted by the authors, and any suggestions from the LLM were critically evaluated before use. The authors bear full responsibility for the research, and the LLM is not listed as a contributor or author.

