# OpenReview forum: "Concise Reasoning in the Lens of Lagrangian Optimization"
_ICLR.cc/2026/Conference — ICLR 2026 Conference Withdrawn Submission_

### Official Review · Reviewer_vuET · 2025-10-28

**Soundness:** 3
**Presentation:** 3
**Contribution:** 3
**Rating:** 6
**Confidence:** 4

**Summary:**

This paper introduces PALU (Performance-Aware Length Update), a framework to achieve concise reasoning in large language models (LLMs). The key idea is to cast conciseness as a constrained optimization problem that minimizes output length under a performance constraint, then solve it using a Lagrangian relaxation. To make this computationally feasible, PALU introduces three approximations: (i) Off-policy performance estimation: reuses old rollouts to estimate pass rate. (ii) Regime-based control: replaces continuous λ updates with a two-regime “bang–bang” rule. (iii) Quantile-driven budget update: uses the dispersion of correct response lengths to approximate ∇ₗR, the sensitivity of reward to length. Experiments show 65% reduction in reasoning length with +15% accuracy improvement across MATH, AIME, AMC, MINERVA, and OlympiadBench tasks on DeepSeek-Distill-Qwen models (1.5B–14B).

**Strengths:**

1. Framing concise reasoning as a Lagrangian optimization problem is novel. While previous studies primarily focus on optimizing accuracy with a length penalty (i.e., $r = a - l$), this paper instead optimizes length with an accuracy penalty applied when performance falls below a predefined threshold c, i.e., $r = l + λ(c - a)$.

2. The three proposed approximations (off-policy, regime-based, and quantile-driven) make the method practically feasible. For instance, the “bang–bang” controller cleverly mitigates instability issues that typically arise from continuous updates of the Lagrange multiplier $\lambda$.

3. The authors conduct thorough experiments across multiple benchmarks, model sizes, and baselines. The consistent performance improvements clearly demonstrate the effectiveness of their proposed method.

**Weaknesses:**

1. The hyperparameter C plays a central role in PALU, yet its selection can critically affect model performance across datasets of varying difficulty. For example, the reasoning models evaluated in this paper typically achieve around 90% accuracy on GSM8K; setting C = 0.8 in such cases could allow PALU to overly relax the performance constraint, leading to a noticeable drop in accuracy. Moreover, the paper does not clearly analyze how different values of C influence the trade-off between conciseness and performance. In practical applications, the difficulty level of the evaluation dataset is often unknown, making it challenging to set C appropriately. The authors should conduct a systematic sensitivity study on C to validate its robustness and practical applicability.

2. Although the authors acknowledge that PALU may perform suboptimally when output lengths concentrate tightly, and they argue that such concentration is rarely observed in practice, there appears to be a clear trend toward tighter output distributions as model size increases. For instance, the 1.5B model produces outputs ranging from approximately 3,000 to 11,000 tokens, whereas the 32B model’s outputs span only 3,000 to 5,000 tokens. This suggests that larger models inherently generate more uniform reasoning lengths, potentially weakening PALU’s effectiveness in larger models.

**Questions:**

1. Can you provide the analysis on $C$?

---

> ### Author Response · Authors · 2025-11-23
>
> We thank the reviewer for the thoughtful feedback. We have revised the paper accordingly and outlined our responses below. We would be grateful to know if these revisions address the reviewer’s concerns.
>
> **W1.1 Role of the performance threshold C in PALU**
>
> $C$ should be modified based on model’s pass rate on *training* samples. In the case of GSM8K as the training dataset, setting C=0.8 could indeed over-relax the performance constraint and lower accuracy. For such easy datasets, a larger threshold, C=1.0, is preferred to preserve accuracy.
>
> **W1.2 Use of C during training or evaluation**
>
> The performance threshold C and the generation budget L are **only active during training**.
> During evaluation or deployment, no *Lagrangian* optimization is performed; all evaluation prompts share a fixed maximum generation length (32k in our case), and  $C$ does not need to be set. Thus, the learned model generates concise outputs under a sufficiently large budget.
>
> **W1.3 Sensitivity and robustness analysis of C**
>
> We add an ablation study sweeping $C\in\{0.6, 0.8, 1.0\}$, as shown in **Figure 4 (right).** The base model’s pass rate on the (math subset of the) training dataset is approximately 0.475. We have the following observation:
>
> - All configurations substantially reduced output length: from 5,700 tokens (the base model) to between 4,200 and 3,012 tokens.
> - $C=0.6$ emphasizes recovery, achieving slightly higher accuracy but with longer outputs (≈770 tokens longer than $C=1.0$).
> - $C=0.8$ achieves a trade-off between conciseness and accuracy.
>
> This study demonstrates PALU’s robustness to different $C$ values and we recommend a lower C value for the following situations: (1) the dataset is very difficult for the base model, (2) the goal of training lies more on concise reasoning rather than accuracy.
>
> **W2.1 On output length uniformity in larger models**
>
> We appreciate your sharp observation. However, the mentioned example is from Figure 1, which illustrates a single prompt. For a broader view, **Appendix Figure 6** shows the response length distributions across multiple prompts. Within the DeepSeek-R1-Distill-Qwen series, larger models indeed generate shorter responses, but this pattern does **not** hold for other families (e.g., Qwen3 or DeepSeek-R1-0528-671B).
>
> We acknowledge that PALU would be less effective for models whose outputs are uniformly long but lack variance in reasoning length. However, our empirical investigation (Appendix Figure 6) indicates that this rarely occurs.
>
> **Q1 Missing analysis on threshold $C$.**
>
> Please refer to Section W1.1 and W1.3. We include a systematic sensitivity study on $C$ demonstrating its effect on both accuracy and conciseness.
>
> **In closing**, we sincerely thank the reviewer for the insightful comments. Addressing these concerns has helped us clarify the role of the performance threshold C in PALU and improve the overall clarity of this paper.

---

> > ### Comment · Reviewer_vuET · 2025-11-25
> >
> > Thank you for your response. I have no further questions.

---

> > > ### Author Response · Authors · 2025-11-26
> > >
> > > Thank you for letting us know that your concerns have been addressed. We are grateful for the reviewer’s careful reading and constructive feedback.

---

### Official Review · Reviewer_dtWr · 2025-10-31

**Soundness:** 3
**Presentation:** 4
**Contribution:** 3
**Rating:** 4
**Confidence:** 4

**Summary:**

The paper proposes Performance-Aware Length Updating, a method to make LLM reasoning more concise and efficient. It formulates concise reasoning as a constrained optimization problem, minimizing reasoning length while maintaining accuracy above a target threshold, using a Lagrangian optimization framework. To make this practical, the authors introduce three approximations: off-policy performance estimation, a regime-based controller for dynamic budget adjustment, and a quantile-driven update rule that adapts token budgets based on the length distribution of correct responses. Evaluation shows improvements in both efficiency and accuracy across multiple reasoning benchmarks and model sizes.

**Strengths:**

- The paper tackles a timely and important problem in LLM research.

- The paper provides a clear theoretical formulation of concise reasoning through a Lagrangian optimization framework, giving a principled view of how to balance accuracy and reasoning length.

- The background and positioning are well done. The paper clearly classifies existing approaches (e.g., fixed budget, reward shaping, pruning-based) and situates PALU as a more general, optimization-driven alternative.

- The assumptions and approximations are well-motivated and intuitively appealing: (i) using an off-policy pass-rate estimate to avoid redundant rollouts, (ii) employing a regime-based controller that dynamically shifts between accuracy recovery and compression, and (iii) adopting a quantile-based surrogate to stabilize gradient updates. Together, these make the framework practical and computationally efficient.

- It provides convincing empirical support for the method's design by including ablation studies that isolate the impact of each approximation.

- The method shows  improvements in both efficiency and accuracy, demonstrating that shorter reasoning traces can indeed be achieved without sacrificing correctness.

**Weaknesses:**

- The evaluation is limited. Experiments are conducted using only one model family (DeepSeek-R1-Distill-Qwen-1.5B), which restricts the generality of the conclusions. It is unclear whether the proposed framework scales similarly to larger or different architectures.


- The computational overhead of the proposed approach is not discussed. Since PALU involves additional monitoring, budget updates, and off-policy estimation, it is important to quantify the real end-to-end runtime or memory cost to validate the claimed efficiency gains.



- The impact of core assumptions and approximations on performance is not analyzed in depth. How sensitive is PALU to these design choices, and could alternative relaxations yield better trade-offs?


- Building on the previous point, there is no systematic ablation to isolate the contribution of each of the three components. Showing how much each assumption contributes to the overall gain (or whether some are redundant) would make the framework more interpretable and robust.


- The paper does not compare alternative theoretical relaxations or optimization formulations. Since the method is inspired by Lagrangian optimization, it would be useful to understand whether simpler or more direct approaches could achieve similar results.



- Although the method is theoretically motivated, the paper lacks deeper intuition or discussion of failure cases. For example, when the quantile update might be unstable or when the off-policy pass-rate estimate becomes inaccurate.

**Questions:**

See the weaknesses section.

---

> ### Author Response · Authors · 2025-11-23
>
> We appreciate that the reviewer’s feedback touches on both the **methodological foundations** and the **evaluation setup**, raising important points about scalability, computational overhead, and robustness analyses. We have revised the paper accordingly and detail our responses below:
>
> **W1 Generality to larger or different architectures.**
>
> We already evaluate PALU on larger models (1.5B, 7B, 14B) and now add results on a different architecture (phi-4-mini-reasoning, 4B).
>
> - In the initial submission, we compared different concise reasoning methods across multiple model scales (1.5B, 7B, and 14B) as shown in Figure 3. This large-scale experiment required approximately **8200 H200 GPU hours**.
> - To further address concerns about generality across architectures, we additionally evaluate PALU on **phi-4-mini-reasoning (4B)** and report the results against two representative baselines in Table 3. PALU consistently maintains its conciseness, confirming its scalability and adaptability.
>
>
> **W2 Missing analysis on computational cost**
> PALU adds ~144 KB of memory overhead and achieves an 8% reduction in end-to-end runtime compared to GRPO. Specifically,
>
> - **(Memory)** PALU introduces three lightweight computations per epoch compared to the GRPO baseline: (1) computing the off-policy pass rate, (2) computing the length quantile, and (3) assigning the length budget for each prompt. For training on a 12k-sample dataset, these operations require storing three arrays of 12k float values, approximately **144 KB** of additional memory. This is negligible compared to model parameters (1.5B to 14B).
> - **(Computation)** Computations of these three arrays occur once per epoch (typically 10–20 epochs), adding an imperceptible cost relative to optimizing a 1.5B-14B parameter model.
> - **(Runtime comparison)** To assess the overall runtime impact, we compare PALU and GRPO under identical conditions (DeepSeek-R1-Distill-Qwen-1.5B, 12k training samples, 20 epochs, two H200 nodes). Both runs include identical evaluation and logging steps. **PALU reduces total wall-clock time by 8.7%**, consistent with its design:
>     - PALU progressively encourages shorter generations during training.
>     - By assigning budgets before rollouts and sorting prompts with similar budgets, PALU forms balanced batches and minimizes idle GPU time.
>
>     We have added a **wall time comparison (Figure 5)** showing reduced rollout time per epoch and summarize the runtime statistics in the following table:
>
> | Method | H200 GPU Hours (2 nodes) | Wall time / epoch  |
> | --- | --- | --- |
> | GRPO (fixed length budget, L = 16k) | 1220 | 3.81 h |
> | PALU (adaptive length budget) | 1113 | 3.48 h |
>
>
> **W3.1 Analysis on the core assumption: the length variance in overthinking trajectories.**
>
> We have already discussed this limitation on **Page 5**, noting that PALU relies on variation in response lengths. Specifically, PALU may not improve conciseness for models that consistently produce verbose responses with identical or narrowly distributed lengths. In this edge case, PALU degenerates to GRPO.
>
> However, this is a **very mild assumption** in practice. We provided an empirical analysis of the generation length distribution per prompt in Figure 6 (Appendix, Page 16), covering the DeepSeek-R1-Distill-Qwen, Qwen3, and DeepSeek-R1 families. The results show that most prompts exhibit broad length distributions, **confirming that the assumption holds for typical reasoning models and does not limit PALU’s applicability.**
>
>
> **W3.2 Impact of the three approximations.**
>
> The three pragmatic approximations translate the theoretical *Lagrangian* optimization into a practical training algorithm. Specifically,
>
> - **Off-policy performance estimation** avoids an other round of rollout (which nearly double the computation cost)**.** To support its reliability, we add an empirical evidence in **Figure 10 (Page 18).** The gap between consecutive epochs’ accuracy (blue points) decreases steadily during training. This verifies that it is reasonable to track a model’s pass rate using its previous records.
> - **Regime-based controller (replacing the Lagrange multiplier)** stabilizes the balance between conciseness and performance in only 10–20 updates (once per epoch). A conventional Lagrange multiplier typically requires **thousands of updates** to converge and an additional learning rate to tune.
> - **Quantile-based length update** replaces the Jacobian term $\nabla_L R(\theta, L, q)$, whose computation is **infeasible in practice.** To see the advantage of our quantile-based approximation, we replaced it with a simple fixed step size (e.g., updating L=L−2048 when performance exceeds C=0.8) and compare them in the middle panel of **Figure 4.** The quantile-based update achieves faster convergence and a better balance between accuracy and conciseness. In contrast, using a fixed step size causes the model to oscillate between improving performance and maintaining conciseness.

---

> ### Author Response · Authors · 2025-11-23
>
> **W3.3 Could alternative approximations yield better trade-offs?**
>
> Other approximations could be possible, but designing and validating them would require a separate study. In this work, we do not claim the proposed approximations are the only or the best possible solution for converting the *Lagrangian* gradient updates into pragmatic implementation.
>
> **W4 Systematic ablation on each of the three approximations**
>
> The new results in Figure 10 support our first approximation. The new ablation in Figure 4 investigates the advantage of the third approximation. For detailed discussion about these new results, please kindly refer to our response to W3.2.
>
> **W5 Compare to alternative theoretical relaxations, and optimization formulations other than Lagrangian**
>
> We thank the reviewer for this suggestion.
>
> We have already compared PALU with **14 competitive baselines** covering both heuristic and reward-shaping based methods, and found that our *Lagrangian*-inspired formulation achieves the best balance between accuracy and conciseness. Exploring alternative theoretical relaxations remains an interesting avenue for future research, but we believe our formulation provides a principled and practically viable foundation.
>
> **W6.1 What if the quantile update (the step size for L) being unstable**
>
> The quantile statistics becomes unstable when the number of rollouts per prompt (N) is very small. However, N is a hyperparameter inherited from **GRPO**, not introduced by PALU. In practice, GRPO itself performs better with larger N (typically N=8 or N=16), which naturally stabilizes PALU’s quantile updates as well.
>
> **W6.2 What if the off-policy pass-rate estimate becomes inaccurate**
>
> The off-policy estimate approximates performance using the previous epoch’s pass rate. It may become inaccurate when (1) the model’s behavior changes drastically between epochs, or (2) the rollout number N is very small.
>
> - For case (1), large behavior shifts are unlikely in our setting: we train on a 12k-sample dataset and observe only minor pass-rate changes across epochs (Figure 10, Appendix, Page 18).
>
> - For case (2), using a sufficiently large N (e.g., N=8) ensures stable estimation and aligns with standard GRPO practice.
>
>
> **In closing**, addressing the reviewer’s concerns has helped us strengthen the soundness, empirical credibility, and interpretability of PALU. We hope that the revised work offers the community a principled, machine-learning-based perspective on concise reasoning in LLMs, beyond heuristic approaches, and we look forward to the reviewer’s feedback.

---

> ### Author Response · Authors · 2025-11-26
>
> We are very grateful for the reviewer’s thoughtful feedback, which has helped us substantially improve the paper. In our revisions, we have (i) expanded the evaluation to an additional model family (phi-4-mini-reasoning), (ii) added a detailed computational cost analysis with a comparison to the GRPO baseline, (iii) incorporate new ablations on the performance threshold C and the step size (against the native fixed step size), and (iv) provided additional empirical evidence supporting the policy pass-rate estimate.
>
> If these updates address the reviewer’s concerns, we would truly appreciate a brief note to the AC confirming this. It would help us ensure that we have fully resolved all issues and allow us to address any remaining concerns as needed.

---

### Official Review · Reviewer_m9Lb · 2025-11-01

**Soundness:** 2
**Presentation:** 2
**Contribution:** 2
**Rating:** 2
**Confidence:** 4

**Summary:**

This paper proposed a new RL algorithm named PALU for training LLMs to achieve efficient reasoning while maintaining accuracy. PALU considered the efficient/concise LLM reasoning as a constrained optimization problem, which is to minimize generation length subject to a performance constraint. With the Lagrangian duality, this problem can be converted into an unconstrained min–max problem, and PALU designed 3 pragmatic approximations in this work. Experimental results showed that the proposed PALU can reduce the output token number and maintain or even surpass the baselines.

**Strengths:**

1. This work considered concise reasoning (efficient reasoning) as a constrained optimization problem with a performance constraint, which is theoretically sound and can align with classical optimization frameworks. Such a perspective provided a clear objective that balances conciseness and correctness, which is better than existing heuristic reward methods.

2. The macro average accuracy of the proposed PLAU was improved, and the token budgets have been decreased to a large margin compared to most baselines.

**Weaknesses:**

1. The presented PALU estimated the expected reward $R(\theta, L, q)$ for the current policy $\pi_{\text{new}} $ using rollouts from the previous policy $\pi_{\text{old}}$, which is a conservative estimate but without quantifying the direction or magnitude of the bias, nor provide empirical analysis of the bias-variance trade-off.

2. When training enters the strong coupling between performance and length regime, i.e., the green phase in Figure 2, rollouts from the old policy may overestimate the new policy’s performance, leading to underestimation of $\lambda$ and excessive length compression. Therefore, the monotonically decreasing curve may be an artifact of this biased estimator.

3. Reducing the continuous $\lambda$ to a binary switch converted the optimization into a discrete. And this paper didn’t provide Lyapunov function, fixed-point analysis, or periodic trajectory proofs, and didn’t not address the possibility of limit-cycle oscillations between the token length and performance.

4. The main experimental results exhibited a better macro average accuracy, but the improvements were mostly from the AMC 2023, while PALU didn’t outperform most baselines on other benchmarks. Also, there is no evaluation on the newest AIME 2025.

5. The Pass@1 is computed over only 10 rollouts (32 for AIME 2024), which may be insufficient for a stable estimation, especially on harder benchmarks like Olympiad or AIME, undermining the confidence for the evaluation.

**Questions:**

1. This paper estimated the expected reward $R(\theta, L, q)$ for the current policy $\pi_{\text{new}}$ via rollouts from the old policy $\pi_{\text{old}}$. When training enters the strong performance–length coupling regime, would this bias always be positive?

2. Could this discrete-event system exhibit limit cycles or chaotic trajectories?

3. When correct samples are extremely scarce (at the early RL stage), how to prevent the step size from exploding or vanishing?

4. How sensitive is PALU to the C threshold? The paper adopted C = 0.8; what happens if C is larger or smaller?

5. All experiments were conducted with DeepSeek-R1-Distilled-Qwen models, have the authors tried other reasoning LLMs?

6. Is the macro average accuracy improvement due to denoising or regularization?

---

> ### Author Response · Authors · 2025-11-23
>
> We appreciate that the reviewer’s feedback touches on both the theoretical formulation and the experimental validation. We have revised the paper accordingly and detail our responses below:
>
> **W1 Bias of the Off-policy performance estimate**
>
> The direction of this bias is already included (Section 4.2) and an empirical analysis is added.
>
> - **(Direction of the bias)** The off-policy performance estimate is a **conservative** estimate. In other words, it underestimates the performance and drives the optimization slightly more on improving the accuracy performance. The detailed discussion can be found in Section 4.2.
> - **(Evidence of the conservative bias)** We add an empirical investigation, Figure 10 in Appendix Page 18, to support out discussion. We plot the evolution of the pass rate on the training data. Two conclusions: (1) The steady evolution of the accuracy supports our using of an old model to approximate the performance of current model. (2) The overall upward trend of the pass rate echoes our justification about the bias (conservativeness) of the off-policy performance estimate.
> - **(What happens with an inaccurate estimate?)** The off-policy performance estimate becomes inaccurate when (1) the model’s behavior changes drastically between epochs, or (2) the rollout number $N$ is very small.
>     - For case (1), large behavior shifts are unlikely in our setting: we train on a 12k-sample dataset and observe only minor pass-rate changes across epochs (Figure 10, Appendix, Page 18).
>     - For case (2), using a sufficiently large $N$ (e.g., N=8 or 16) ensures stable estimation and aligns with standard GRPO practice (N is a hyper-parameter in GRPO).
>
> **W2 The monotonically decreasing in generation length seems an artifact of this biased estimator**
>
> This is a great question, and we appreciate the opportunity to address it, as it helps improve the soundness of our work.
>
> - **Figure 2** presents the evolution of accuracy on the **evaluation set**, while the corresponding trend on the **training set** is shown in **Figure 10 (Appendix)**. We believe it is more rigorous to analyze the estimator bias on the training set, as this bias directly affects the training dynamics.
> - In **Figure 10**, the **gap** between consecutive epochs’ accuracies (blue points) steadily decreases throughout training, providing evidence against the conjecture that PALU consistently and significantly overestimates accuracy.
> - To further address this concern, note that if the off-policy performance estimate were to significantly overestimates accuracy, it would trigger an **over-reduction of the length budget L**, causing a drop in training accuracy. However, in the **next epoch**, PALU resets L to its maximum value.
> - We also empirically confirm that **overly aggressive manual reductions**, such as the stage-based scheduling from 16 k to 8 k over five stages (Figure 3), result in **degraded performance**, validating the importance of PALU’s adaptive length updating.
>
> **W3 Lyapunov function and fixed-point analysis for a LLM system**
>
> We do not view the absence of a Lyapunov or fixed-point analysis as a weakness. Formal Lyapunov or fixed-point analyses are uncommon in alignment or reasoning. Following the standard practice in GRPO, we focused on demonstrating stable convergence through empirical results (e.g., Figure 2 and Figure 3). They consistently demonstrate smooth and stable convergence in practice.
>
>
> **W4.1 Improvements on accuracy mainly from AMC 2023**
>
> We respectfully disagree that this represents a weakness.
>
> PALU is designed to promote *concise reasoning.* We therefore see the generation length reduction and the AE score, which jointly measures performance and conciseness, better reflects its goal. PALU outperforms in terms of the average AE score(Table 2) and per-task AE score (Table 7, Appendix, p. 15).
>
> **W4.3 Evaluation results on AIME 2025**
>
> Added in Table 2, Table 3 and Table 10.
>
> **W5  Rollout number for evaluation**
>
> **There is a trade-off between evaluation stability and computational cost**. While increasing the rollout number can yield a more stable estimate, it also incurs significant compute overhead. In our setting, even with 10 rollouts for large datasets and 32 for smaller ones (e.g., AIME 2024), each single-row result in Table 1 requires about **24 GPU hours** (H200). Considering the **14 baselines compared**, larger rollout numbers would be prohibitively expensive.
>
> Our evaluation protocol follows [1], ensuring a fair and standard comparison under reasonable computational budgets. For reference, [2] reports results in 3 trails and [3] use 4-64 rollouts according to the size of the evaluation dataset.
>
> - [1] The Overthinker's DIET: Cutting Token Calories with DIfficulty-AwarE Training
>
> - [2] Phi-4-Mini-Reasoning: Exploring the Limits of Small Reasoning Language Models in Math
>
> - [3] DeepSeek-R1: Incentivizing Reasoning Capability in LLMs via Reinforcement Learning

---

> ### Author Response · Authors · 2025-11-23
>
> **Q1 Bias of the Off-policy performance estimate**
>
> Discussed in our response to W1.
>
> **Q2 Could PALU exhibit limit cycles or chaotic trajectories?**
>
> The first author of this paper lacks the knowledge for understanding or answering this question. She/He respectfully refers the reviewer to Figures 2 and 3 for empirical evidence on PALU’s training dynamics. If the question concerns a different aspect, we would appreciate it if the reviewer could kindly cite relevant paper (preferably within the LLM literature) for reference.
>
> **Q3 How to prevent the step size from exploding or vanishing when correct samples are extremely scarce?**
>
> PALU does not compute the step size in that case. (This question touches on a key design choice in PALU! Thanks!)
>
> - When correct samples are extremely scarce, PALU directly resets the length budget to a large value, $L = L_\max$, since the performance threshold is not satisfied.
> - The relevant third approximation used in PALU, $\nabla_L R(\theta, q, L) \propto \alpha_\tau$ is only valid when correct samples are sufficient (R≥C). See the derivation for Eq 15.
>
> **Q4 What happens if threshold C is larger or smaller?**
>
> We add an ablation study sweeping $C\in\{0.6, 0.8, 1.0\}$, as shown in **Figure 4 (right).** The base model’s pass rate on the (math subset) training dataset is approximately 0.475. We have the following observation:
>
> - All configurations substantially reduced output length: from 5,700 tokens (the base model) to between 4,200 and 3,012 tokens.
> - $C=0.6$ emphasizes recovery, achieving slightly higher accuracy but with longer outputs (≈770 tokens longer than $C=1.0$).
> - $C=0.8$ achieves a trade-off between conciseness and accuracy.
>
> This study demonstrates PALU’s robustness to different $C$ values and we recommend a lower C value for the following situations: (1) the dataset is very difficult for the base model, (2) the goal of training lies more on concise reasoning rather than accuracy.
>
> **Q5 Results on other reasoning LLMs**
>
> Added in **Table 3**.
>
> We provided comparison on Phi-4-mini-reasoning, a 4-B reasoning model **in addition to DeepSeek-R1-Distill-Qwen-1.5, 7B, and 14B models**. The results are reported in Table 3. Owing to the computational cost of each run, we include: ThinkPrune-4k, and the overlong-punishment method proposed by DAPO. PALU attains comparable accuracy to the base model while maintaining a substantially shorter generation length, in contrast to alternative methods that sacrifice accuracy for brevity.
>
> **Q6 Is the macro average accuracy improvement due to denoising or regularization?**
>
> PALU outperforms existing heuristic approaches. Among them, *ShorterBetter* achieves the greatest reduction in generation length; however, this “**denoising**” alone does not yield accuracy gains in their case, suggesting that **Occam’s razor** does not apply.
>
> We attribute PALU’s improvement in average accuracy to its **Lagrangian formulation**. Specifically, the **Lagrange multiplier term,** which may resemble a regularization term, **explicitly balances** maintaining performance and reducing the generation budget, thereby enabling PALU to achieve both concise and accurate outputs.
>
> **Finally**, we would like to thank the reviewer for their time and consideration and look forward to the forthcoming feedback.

---

### Official Review · Reviewer_C3Ax · 2025-11-01

**Soundness:** 4
**Presentation:** 3
**Contribution:** 4
**Rating:** 8
**Confidence:** 2

**Summary:**

Concise reasoning cab seen as constrained optimization! Simply the authors frame it as: minimize response length L subject to performance R(theta,L,q) >= C. Converts to Lagrangian min-max problem L(theta,L,λ) = L + λ(C - R(theta,L,q)) with updates λ <- λ + n(C-R), theta ← theta + nλ∇R, L <- L - n(1-λ∇R). Proposes PALU with three approximations: (i) off-policy performance estimation reuses last epoch's rollouts; (ii) regime-based optimization snaps λ to two extremes - if R>=C reduce L by α_τ^q, else reset L=L_max; (iii) quantile-driven budget update where α_τ^q = Q_1.0^q - Q_{1-τ}^q measures gap between longest correct response and (1-τ) quantile, approximating ∇R via finite difference τ/α_τ^q.

**Strengths:**

I love this paper. Because it has a principled formulation. Grounding concise reasoning in Lagrangian optimization is elegant and provides theoretical justification for balancing length/accuracy unlike ad-hoc heuristics. Quantile-based step size α_τ^q is clever - using distribution of correct response lengths to determine reduction magnitude is data-driven and intuitive. Strong empirical results - 65% compression with 15% accuracy gain on 1.5B model beats all baselines including multi-stage methods like AutoThink. Figure 2 left shows nice dynamics - negative correlation early (easy redundancies) then positive (hard tradeoffs) with PALU continuing to improve both. Cross-domain and cross-scale experiments (Figure 3) demonstrate genuine adaptivity - stage-based budgeting and overlong punishment fail when initial length differs from assumed 8k target while PALU adapts naturally. AE Score metric properly weights accuracy preservation (theta=5 penalty vs n=3 bonus). Off-policy performance estimation is pragmatic - reusing last epoch avoids expensive recomputation. Generation length assumption validated empirically (Figure 1, Figure 5) showing 2-3x spread supporting premise.

**Weaknesses:**

Regime-based controller might be too simplistic? - binary snap to L_max or reduce by α_τ discards continuous λ updates that Lagrangian theory prescribes. There is a claim this "preserves sign behavior" but loses magnitude information that could enable finer control. Why not just use gradient ascent with proper learning rates? Off-policy approximation is conservative but introduces lag - performance might improve but policy still thinks constraint violated leading to unnecessary L_max resets? I would love to see an analysis of how often this happens. Quantile approximation  assumes reducing L by α_τ drops success rate by exactly τ but this only holds if L is near Q_1.0. For smaller L values this approximation breaks down? Figure 4 ablation shows α_0.1 vs α_0.5 but doesn't test different τ values in quantile definition. Performance threshold C=0.8 fixed across all experiments - why 0.8? Compute cost comparison missing - paper claims "efficiency" but reports 1100 H200 GPU hours for 1.5B model which seems expensive. How does this compare to baseline GRPO training?

**Questions:**

How does PALU behave when performance constraint C cannot be satisfied? On AIME24 baseline is 28.5% so initial performance is far below C=0.8. Does L just stay at L_max indefinitely? Table 2 shows final AIME24 accuracy 40.0% still below threshold - what is the training dynamic here? Can you provide analysis of how often regime controller resets to L_max vs reduces by α_τ? This would show whether binary controller actually captures Lagrangian dynamics or just oscillates. The quantile gap α_τ^q requires computing Q_1.0^q which is the longest correct response - but if some questions have very long outlier solutions this could make α_τ^q very large causing aggressive reduction. Do you clip or filter outliers? Why τ=0.5 specifically?

---

> ### Author Response · Authors · 2025-11-23
>
> Thank you for your thoughtful feedback on our work. We have revised the paper accordingly and detail our responses below. Please let us know if these address your concerns.
>
> **W1 Regime-based controller v.s. gradient ascent on $\lambda$.**
>
> In our setting, each prompt affords only 10–20 update steps, whereas continuous gradient ascent on the *Lagrangian* multiplier $\lambda$ typically requires thousands of iterations to converge. We therefore use a sign-based controller that switches between focusing on accuracy or conciseness. This discrete regime effectively balances accuracy and length as evaluated, while exploring continuous $\lambda$ updates remains a promising direction for future work.
>
>
> **W2 The lag in off-policy performance estimate leads to unnecessary $L_\max$ reset.**
>
> This is a great question, thanks!
>
> - To quantify this lag, **Figure 10 (Appendix p. 18)** plots the evolution of the pass rate across epochs. The gap between consecutive epochs, in other words, the lag, decreases steadily during training. Thus unnecessary $L_\max$ resets caused by the lag do not dominate the $L$ update.
> - Although directly computing how often the lag causes resets would require costly on-policy rollouts, the evolution of the length budget $L$ provides indirect evidence (**Figure 10 (Appendix p. 18)** ). As training proceeds, PALU assigns more samples with reduced budgets, and by the final epoch, 5 out of 24 rollout batches use $L < L_\max$.
>
> **W3 Quantile approximation breaks with small $L$.**
>
> This issue does not arise in our setting.
>
> - Recall that $L$ denotes the generation budget from the previous epoch; $Q_{\tau=1.0}$ is the longest correct response. Because correct responses must fit within the current budget, we always have $L \geq Q_{\tau=1.0}$. In other words, small $L$ happens with smaller $Q_{\tau=1.0}$.
> - Our quantile approximation adaptively scales with the observed response lengths. ****To see this**,** consider updating $L$ with a fixed-step update , e.g., $L = L - 4096$ if $R>C$. When $L$ is small, the estimated quantile gap becomes correspondingly small, ensuring stable and smooth updates. While the fixed-step update fails.
>
> **W4 Different $\tau$ values in quantile definition.**
>
> The quantile approximation term is defined as $\alpha_\tau = Q_{1.0} - Q_{\tau}$. We focus on the Jacobian near the longest length, i.e., $L= Q_{\tau=1.0}$. In the ablation study (the left panel of Figure 4), $\alpha_{\tau=0.5}$ denotes using $Q_{\tau=1.0} - Q_{\tau=0.5}$ as the step size for updating $L$.
>
> **W5 Ablation on performance threshold $C$**
>
> We set $C=0.8$ without extensive tuning. A lower threshold, e.g., $C=0.6$, emphasizes conciseness but slightly reduces accuracy. This trade-off is confirmed by the ablation study on $C$ in the right panel of Figure 4.
>
> **W6 Compute cost comparison**
>
> We compare PALU and GRPO under identical conditions (DeepSeek-R1-Distill-Qwen-1.5B, 12k training samples, 20 epochs, two H200 nodes). Both runs include identical evaluation and logging steps. **PALU reduces total wall-clock time by 8.7%**, consistent with its design:
>
> - PALU progressively encourages shorter generations during training.
> - By assigning budgets before rollouts and sorting prompts with similar budgets, PALU forms balanced batches and minimizes idle GPU time.
>
> We have added a **wall time comparison (Figure 5)** showing reduced rollout time per epoch and summarize the runtime statistics in the following table:
>
> | Method | H200 GPU Hours (2 nodes) | Wall time / epoch  |
> | --- | --- | --- |
> | GRPO (fixed length budget, L = 16k) | 1220 | 3.81 h |
> | PALU (adaptive length budget) | 1113 | 3.48 h |
>
> The description “efficiency” refers to feature that we use off-policy performance estimate to compute the very expensive on-policy accuracy.

---

> ### Author Response · Authors · 2025-11-23
>
> **Q1 What if performance constraint C cannot be satisfied on difficult tasks such as AIME 24?**
>
> The performance threshold C and the generation budget L are **only active during training**. During evaluation, no *Lagrangian* optimization is performed; all evaluation prompts share a fixed maximum generation length (32k in our case). Thus, the learned model generates concise outputs under a sufficiently large budget, and $C$ does not need to be set at inference time.
>
> **Q2 How often regime controller resets to L_max v.s. reduces by α_τ?**
>
> In the final epoch of the run shown in Figure 10 (Appendix p. 18), 5 out of 24 rollout batches use $L<L_\max$, while the remaining 19 batches reset to $L_\max$. Each batch has 512 prompts.
>
> **Q3 What if the step size $\alpha_{\tau}^q$ is very large?**
>
> For example, suppose the eight responses for a given prompt are all correct, with one being 16k tokens long and the others under 1k. In this case, PALU, with the help of **$\alpha_{\tau}^q$,** reduces the length budget $L$ directly to around 1k, covering roughly 50% of the correct responses ($\tau = 0.5$). In contrast, a fixed-step method would require multiple epochs to reach the same level.
>
> This seemingly “aggressive” reduction does not harm performance, since the resulting accuracy remains near $\tau = 0.5$ (a hyper-parameter). The advantage of using a quantile gap over a fixed step size (e.g., L=L-2048) is empirically validated in the middle panel of Figure 4.
>
> **Q4 Why $\tau=0.5$ specifically?**
>
> We conducted two runs with $C = 0.8$ and $\tau \in {0.8, 0.5}$. The setting $\tau = 0.5$ achieved slightly better conciseness, so we report results with it. For a direct comparison across $\tau$ values, please refer to the left panel of Figure 4.

---

### Author Response · Authors · 2025-11-30
**Concise Summary of Rebuttal Responses**

Dear Area Chair, Reviewers, and Readers,

The authors thank the Area Chair for the time and service to the community, especially amid a heavy reviewing load this year, and appreciate the reviewers’ thoughtful feedback. A concise summary of the rebuttal is provided below for reference.

| **Concerns & Suggestions (Method)** | **By Reviewer** | **Our Action** |
| --- | --- | --- |
| The lag in off-policy performance estimate may make PALU unstable. | **C3Ax (W2), m9Lb (W1), dtWr (W6.2)** | Referenced **Figure 10** showing the lag decreases steadily during training, and clarified that an ideal on-policy estimate would nearly double the training cost. |
| Regime-based controller v.s. gradient ascent on the *Lagrange* multiplier. | **C3Ax (W1)** | Explained that continuous λ-updates are impractical with only 10–20 update steps. |
| What if the quantile approximation (for updating the length budget L) breaks? | **C3Ax (W3), dtWr (W6.1)** | Clarified that quantile accuracy depends on GRPO’s $N$, number of rollouts for each prompt, and that the reviewer’s concern does not arise in our setting. |
| Assumption on the output length uniformity in LLMs | **vuET (W2), dtWr (W3.1)** | Referenced **Figure 6** showing the length-variation assumption holds reliably in LLMs. |
| Impact of the three approximations. | **dtWr (W3.2)** | ***(1, Evidence)*** Validated the off-policy estimate using empirical evidence (**Figure 10**). **(*2, Discussion)*** Explained why continuous λ-updates are impractical. ***(3. Comparison/Ablation)*** Compared quantile-based vs. fixed step sizes (**Figure 4**). |
| Missing Lyapunov function and fixed-point analysis. | **m9Lb (W3)** | Clarified that such analyses are **uncommon and infeasible** for LLMs; referred to **Figures 2–3** for PALU’s convergence behavior. |
| Comparison with alternative relaxations/formulations (beyond Lagrangian). | **dtWr (W5)** | Clarified that introducing the *Lagrangian* formulation is a key novelty; exploring other formulations is **beyond scope**. |
| Comparison with alternative approximations. | **dtWr (W3.3)** | Clarified that our three approximations make *Lagrangian* optimization practical; evaluating all other potential approximations is **beyond scope**. |
****

| **Concerns & Suggestions (Evaluation)** | **By** | **Our Action** |
| --- | --- | --- |
| Generality to larger or different architectures. | **dtWr (W1)** | Already evaluated PALU on 1.5B/7B/14B models; added results on a different architecture (**phi-4-mini-reasoning, 4B**) in **Table 3**. |
| The monotonically decreasing in generation length (Figure 2, Left) seems an artifact of the biased off-policy performance estimate.  | **m9Lb (W2)**  | ***(1)*** Bias should be evaluated on the training set, not validation. ***(2)*** **Figure 10** shows estimator bias steadily decreases. ***(3)*** **Figure 3** shows fixed step sizes (the effect of strong bias) degrade performance, while PALU’s performance-aware length reduction strategy maintain accuracy.
 |
| Ablation study on the performance threshold C in PALU | **vuET (W1), C3Ax (W5)** | Added discussion on the role of **C** and included a sweep in the right panel of **Figure 4**. |
| Different  tau values in quantile definition. | **C3Ax (W4)** | Clarified that multiple tau values were already tested; sweep results included in **Figure 4 (left)**. |
| Systematic ablation on each of the three approximations | **dtWr (W4)** | (***1)*** **Figure 10** supports the off-policy estimator is reliable. ***(2)*** Added discussion explaining why continuous λ-updates are impractical. ***(3)*** **Figure 4** (the middle) evaluates the quantile-based step size approximation against the fixed step size (ablation). |
| Improvements on accuracy mainly from one benchmark (AMC 2023). | **m9Lb (W4.1)** | Clarified that PALU’s goal is reducing length while maintaining accuracy, where many baselines fail, rather than improving accuracy. |
| Missing evaluation results on AIME 2025 | **m9Lb (W4.3)** | Added results in **Tables 2**, **3**, and **10**. |
| Insufficient rollout number for evaluation | **m9Lb (W5)**  | Clarified that our evaluation follows community practice; increasing rollouts is computationally expensive (each row ≈24 H200 GPU hours, ×14 baselines). |
| Missing analysis on computational cost | **dtWr (W2), C3Ax (W6)** | Added memory analysis (144 KB overhead) and wall-time comparison vs. GRPO (**Figure 5**). |
****

We have updated the paper with new evidence and clarifications based on the reviewers’ feedback, and we hope the revised version provides a principled, ML-based perspective on concise reasoning in LLMs.


Sincerely,

The Authors of Submission 19020

---

### Note · Authors · 2026-01-28

I have read and agree with the venue's withdrawal policy on behalf of myself and my co-authors.

---

### Meta-Review · Area_Chair_WEDR · 2025-12-27

**Summary:**

This paper proposes PALU (Performance-Aware Length Updating), framing concise reasoning as a constrained optimization problem: minimize generation length subject to maintaining performance above a threshold, then solving via a Lagrangian-inspired approach with practical approximations (off-policy performance estimate, bang–bang/controller-style multiplier behavior, and quantile-based length updates). Reviewers generally agree the formulation is principled and the approach is pragmatic for RL-style post-training, with strong empirical evidence that PALU can substantially reduce reasoning length while maintaining (and sometimes improving) accuracy across multiple benchmarks and model scales. Concerns centered on (i) off-policy estimate bias/lag and stability, (ii) discretizing the multiplier dynamics, (iii) sensitivity to hyperparameters (notably threshold C and quantile parameter tau), and (iv) evaluation breadth and compute/runtime reporting. The rebuttal adds targeted ablations and additional results (incl. compute/runtime comparison to GRPO, sensitivity to C, tau sweep, added benchmark results, and at least one additional architecture), which addresses some practical concerns.

**Reviewer Concerns:**

Concerns substantially addressed by rebuttal:

- Compute/runtime and overhead: authors added a wall-time comparison vs. GRPO and reported modest runtime improvements plus small memory overhead; this directly addresses “efficiency claims need quantification.”

- Generality beyond a single setting: added results beyond the initial core setup (larger scales already present; an additional architecture was added in revision), partially mitigating “single-family evaluation” concerns.

- Hyperparameter sensitivity (threshold C, quantile tau): authors added a sweep for C and clarified guidance for easy vs. hard datasets; they also clarified/testing around tau

- Off-policy estimator bias/lag and quantile-update stability: authors provided empirical evidence (lag decreasing over training, stability discussion tied to rollout count N), and clarified failure modes; they also gave some accounting of how often reduced budgets
 happen late in training.

- Missing benchmark coverage (e.g., AIME 2025): authors added the missing evaluations.

Concerns still outstanding:

- Theory/stability of the discrete controller: one reviewer requested deeper dynamical-systems style analysis (Lyapunov/fixed points/limit cycles). The revision argues this is uncommon/infeasible for LLM training and points to empirical stability; however, the direct response to the “limit cycles/chaos” question is weakly handled and does not provide additional insight beyond empirical curves.

- Breadth of ablations: while the revision adds evidence for key approximations and a comparison against fixed-step updates, a fully systematic component-by-component ablation and exploration of alternative relaxations is still limited.

**Reviewer Scores:**

I cannot reliably answer this counterfactual question without putting words in reviewers’ mouths. I will not impute score changes beyond what reviewers explicitly stated in the discussion. I instead provide a faithful synthesis of the discussion outcomes and remaining points of disagreement.

---

### Decision · Program_Chairs · 2026-01-26

Reject